# A slow-fast trait continuum at the whole community level in relation to land-use intensification

Organismal functional strategies form a continuum from slow- to fast-growing organisms, in response to common drivers such as resource availability and disturbance. However, whether there is synchronisation of these strategies at the entire community level is unclear. Here, we combine trait data for >2800 above- and belowground taxa from 14 trophic guilds spanning a disturbance and resource availability gradient in German grasslands. The results indicate that most guilds consistently respond to these drivers through both direct and trophically mediated effects, resulting in a 'slow-fast' axis at the level of the entire community. Using 15 indicators of carbon and nutrient fluxes, biomass production and decomposition, we also show that fast trait communities are associated with faster rates of ecosystem functioning. These findings demonstrate that 'slow' and 'fast' strategies can be manifested at the level of whole communities, opening new avenues of ecosystem-level functional classification.

Understanding how functional strategies respond to environmental drivers is one of the longest-standing and most fundamental questions in ecology (e.g., r/K strategist theory[1], CSR (competitive/stress-resistant/ruderals) strategies[2]). Due to evolutionary trade-offs, species allocate resources differently to their capacity to grow, reproduce, and survive, and for several taxa, it is well established that this leads to sets of co-varying traits that represent ecological strategies[3–5]. At the community level, a range of positive and negative biotic interactions[6] and abiotic factors constrain which individuals, bearing specific traits, persist in a community[7]. In any given environment this is likely to lead to the dominance of the trait set best adapted to local conditions, leading to trait similarity between co-occurring species and resulting in community-level trait covariation.

Functional strategies, and their trait proxies, have been particularly well studied and characterised in vascular plants, both at the species[2,8–10] and community levels[11], but have also been described for groups as diverse as fishes, arthropods[12], and more recently, microorganisms[13]. While the hypotheses underlying such patterns are often underdeveloped compared to those for plants, similar drivers of strategy orientation have been identified, namely resource availability

and disturbance; and these are consistently seen to act concurrently to shape both individual species and community-level strategies traits in terms of growth, reproduction, and survival across the tree of life[14,15]. Body size in particular is a fundamental trait of functional strategies that shows consistent responses to these drivers, with undisturbed environments filtering for larger ('slow') organisms and disturbed environments for smaller ('fast') organisms across groups[14,16], a finding that is consistent with general theory[17].

At the level of guilds and communities, winning strategies can be seen as manifesting as an emergent property, and represented in community-level trait measures, typically the community abundance-weighted trait mean[11,18] (CWM). While there can be significant trait variation within a community, a CWM captures the average functional strategy of the community; and changes in CWM across space and time reflect both turnover in species with different trait values, and variation in their relative abundance, in response to changes to the species pool and environmental conditions. The combined response of multiple trait CWMs thus represents a change in the overall functional strategy at a community level. This means that slow-fast strategy responses – encompassing a range of traits – may emerge at the

✉e-mail: margot.neyret@univ-grenoble-alpes.fr; peter.manning@uib.no

community level from a concurrent change in individual CWM traits related to 'slow' and 'fast' strategies. In particular, communities of resource-acquisitive, fast-growing organisms with numerous off-spring, a fast pace of life and good dispersal abilities tend to be found in resource-rich and disturbed habitats, while resource-conservative, slow-growing organisms with longer lifespan and fewer offspring tend to be favoured in undisturbed or resource-poor habitats[14,15,19–21].

Consistent with these common responses at both species and community levels, associations between traits have been reported between interacting guilds, including plants and soil microorganisms[22–24], plants and arthropods[25], and plants and frugivores[26]. These shared responses between guilds to similar environmental drivers, along with the existence of widespread strategies within guilds, suggest the possibility of synchronised responses across trophic groups and therefore the potential existence of trait syndromes at the level of entire communities. The possibility of such community-level coupling[27] has been discussed previously, in particular in terms of linkages between plants, herbivores and the soil food web[28]. However, while there is theoretical and empirical support for a community-level slow-fast trait response, conflicting observations challenge this hypothesis. First, while the slow-fast spectrum is well defined in plants, additional axes of variation in functional and life-history strategies (e.g., reproductive strategies, and their defining traits such as the timing of reproductive onset and reproductive allocation) have been identified in both plants and other organisms[29,30], and these may respond to different drivers and dominate the distribution of certain organismal groups, decoupling them from the response of others[3,31]. Furthermore, guilds might vary in their strength of response to the drivers of the slow-fast functional axis[32] and could respond over different spatiotemporal scales[33] leading to weak overall coupling.

If community-level slow-fast responses to disturbance and resources are present, then they may be driven by shared direct responses to the environment such as rapid reproduction and effective dispersal of both plants and their consumers in highly disturbed environments. In contrast, synchronous responses could also be mediated by bottom-up[34] trophic interactions: at higher resource availability, we expect higher productivity and resource concentrations, and faster rates of resource transfer between trophic levels, encouraging organisms with faster growth and reproduction both above and belowground[28,35]. Thus, the slow-fast response of the entire community to resource and disturbance drivers might emerge from both direct effects (mainly through shared responses to disturbance) and indirect effects (e.g., a trophic cascade driven by the resource availability component).

Finally, many studies have shown that community-level trait measures of individual guilds explain variation in individual ecosystem functions, which is to be expected given the link between functional traits and rates of metabolic and tissue turnover processes[23,36,37]. Meanwhile, several recent studies have demonstrated that overall ecosystem functioning can be described in terms of just a few fundamental functional axes[38–40]. Given this, we predict that the entire community 'slow-fast' trait axis corresponds to an ecosystem functioning 'slow-fast' axis, with 'fast' functioning defined as fast process rates (e.g., high productivity, rapid decomposition and nutrient turnover).

While theory often considers the effects of resource availability and disturbance as orthogonal (e.g., CSR theory[2]), these drivers are often confounded in real ecosystems. Agricultural systems in particular, tend to be found on a continuum between low intensity (low disturbance, no nutrient input) and high intensity (high disturbance, such as mowing, and high nutrient input). We can thus expect to find a continuum between 'slow and steady' to 'grow fast, die young' strategies along land-use intensity gradients – and indeed land-use intensification tends to select for faster strategies of plants[41], arthropods[25,42], and microorganisms[43].

Despite the evidence base presented above, until now, a lack of coordinated multitrophic abundance, functional trait, and ecosystem function data has hindered the investigation of synchronised responses at the level of entire communities, and their link to ecosystem functioning. However, a recent explosion in trait data availability[44–47], combined with large-scale research platforms that survey multiple organismal groups and ecosystem functions simultaneously[48–50], now allows this long-standing question to be addressed. Here, we use data from the large-scale and long-term Biodiversity Exploratories[49] to test the hypotheses that there is a common, whole community-level slow-fast response to disturbance and resource availability in the form of land-use intensity, and that this community-level strategy drives a slow-fast ecosystem functioning response. The Biodiversity Exploratories is, to our knowledge, the most comprehensive data source for multiple guilds, with abundance data available from the same sites and at all trophic levels, both above- and belowground, including bacteria, fungi, protists, as well as plants, invertebrates and vertebrates[33]. These are collected in 150 grassland plots in three regions of Germany. We focus on the land-use intensity gradient of the Exploratories, which is a combined gradient of both resource availability (fertilisation, with high fertilisation being usually applied to the most inherently productive sites) and disturbance (mowing, which is strongly correlated with fertilisation; and grazing)[51]. Because existing theory on slow-fast strategies was lacking for some guilds, we first conducted expert workshops to identify traits expected to represent the slow-fast continuum for each guild (Tables S1 and S2). Based on pre-defined hypotheses from existing theories, observational studies, and expert knowledge, we selected several traits for each guild to represent the slow-fast continuum, and generated expectations of how these respond to resource availability and disturbance (Fig. 1, see Table S2 for full detail). Focusing on these pre-selected traits, we then tested three hypotheses: (H1) there is a synchronous functional response across trophic levels, shifting from 'slow and steady' to 'grow fast, die young' strategies with increasing land-use intensity. (H2) Land-use intensity drives the slow-fast response of the entire community through both direct and indirect (through a trophic cascade) pathways. (H3) The entire community 'slow-fast' trait axis corresponds to an ecosystem functioning 'slow-fast' axis, with 'fast' functioning defined as fast process rates (e.g., high productivity, rapid nutrient turnover). We show that the slow-fast response is coordinated among most guilds and drives an ecosystem functioning slow-fast axis, thus forming a whole ecosystem slow-fast axis.

## Results

We first identified 'slow-fast' traits at the community level within individual trophic guilds. To do this, we classified individual species into trophic guilds of broadly comparable trophic status (e.g., aboveground primary consumers or belowground omnivores). Different taxonomic groups with the same trophic status were aggregated into a single trophic guild if all trait data was consistent, otherwise they were kept separate. For instance, we had comparable trait data for many aboveground arthropods (size, feeding generalism, dispersal ability compiled from ref. 52, but data for Lepidoptera was available for different traits (hibernation stage, voltinism, etc) and compiled from a range of other sources[53–56]. Therefore, this group was treated separately from other primary consumer arthropods. This led to the definition of 14 homogeneous trophic guilds (hereafter 'guilds'), listed in Table 1. For each guild, abundance-weighted, community-level mean (CWM) trait values were calculated and corrected for environmental covariates (see Methods). Some expected trait responses of different guilds to resource availability and disturbance are shown in Fig. 1 (see Table S2 for full details).

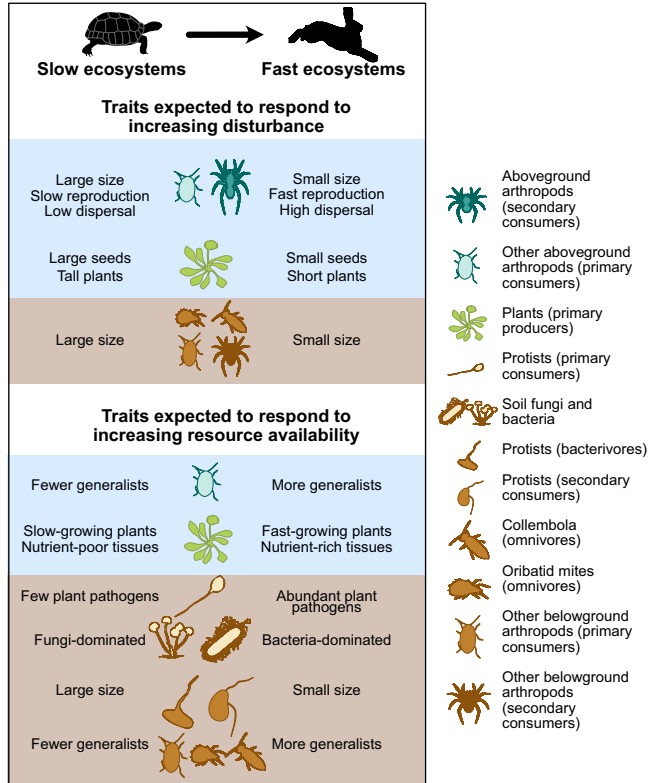

**Fig. 1 | Expected trait variations in 'slow' (low resource, low disturbance) and 'fast' (high resource, high disturbance) communities.** The tortoise and the hare icons represent Aesop's fable "The Tortoise and the Hare", which is the origin of the phrase "slow and steady wins the race", here a potentially winning ecological strategy under conditions of low resource availability and low disturbance. The resource and disturbance gradients are, in this study, respectively represented by a fertilisation and a grazing/mowing gradient. Only a subset of hypotheses is shown[8,15,19,21,22,24,57,61,64,147,268], see Table S2 for full details and references. Belowground guilds are shown in brown, aboveground guilds in blue. Icons were acquired and adapted from Phylopic.org (artists: M. Dahirel, B. Lang, M. Crook, J. A. Venter, H. H. T. Prins, D. A. Balfour, R. Slotow, T. M. Keesey, A. A. Farke, Y. Wong, G. Monger).

## Identification of guild-level slow-fast axes

We assumed that for each guild, the main axis of covariation between the selected slow-fast traits (Fig. 2) should represent the guild's overall slow-fast response ('axis'). If the selected traits were not strongly associated (weak covariation) in a principal coordinates analysis (PCA), we concluded that there was no observable slow-fast axis in the considered guild. Conversely, if most selected traits within a guild covaried, then their main axis of covariation was retained as the guild's slow-fast axis. This covariation axis was defined through a principal component analysis with all traits within each guild, and retained if it explained more than twice of the shared variance as expected if traits were independent; and if the axis was correlated ($r > 0.4$) with at least 60% of the traits (see specific criteria in Methods).

The existence of the hypothesised guild-level slow-fast axis was supported by the data for most guilds, and it explained on average 52% ($\pm$ sd 13) of the total variation in the 4.2 ($\pm$ 1.8) CWM traits per guild that were included in the analyses. For three protist guilds (primary consumers, bacterivores, secondary consumers), only one trait was available, cell size, and we therefore assumed the existence of a 'slow-fast' axis defined by large to small cell size in subsequent analyses[1,57]. Of the remaining 11 trophic guilds, we found strong support for a community-level 'slow-fast' axis in nine guilds: plants (primary producers), aboveground arthropods (primary consumers), aboveground arthropods (secondary consumers), birds (tertiary consumers), belowground bacteria and fungi (decomposers), belowground arthropods (primary consumers), Collembola (omnivores), Oribatid mites (omnivores), and other belowground arthropods (secondary consumers). For instance, aboveground arthropods (primary consumers) displayed a clear trade-off between 'slow' communities dominated by large body size and slow reproduction, and 'fast' communities dominated by generalist-feeding species with smaller body size, higher dispersal abilities and multiple generations per year (PC1: 53.8% variance, Fig. 2). Similarly, Collembola (omnivores) displayed a trade-off between 'slow' communities, dominated by soil-surface-dwelling species with a large body size and sexual reproduction, and 'fast' communities, characterised by more organisms living deeper in the soil and capable of parthenogenesis, which leads to faster generation times (PC1: 65% variance). There was partial support for the existence of a 'slow-fast' axis in the remaining guilds (Fig. 2).

## Table 1 | Trophic guilds taxonomic composition and assigned trophic position

| Guild | Taxa included | Trophic position |
|---|---|---|
| Birds (tertiary consumers) | Insectivorous and predatory birds | 3 |
| Bats (tertiary consumers) | Bats | 3 |
| Herb- and litter-dwelling arthropods (secondary consumers) | Omnivorous Hemiptera, carnivorous Coleoptera, Araneae, Orthoptera and Hemiptera | 2 |
| Lepidoptera (primary consumers) | Butterflies and day-flying moths | 1 |
| Other herb- and litter-dwelling arthropods (primary consumers) | Herbivorous Orthoptera, pollinators, herbivorous and detritivorous Coleoptera, and herbivorous Hemiptera | 1 |
| Vascular plants (primary producers) | All vascular plants | 0 |
| Microbial communities (decomposers) | Bacteria and fungi, including saprotrophs and parasites | 1 |
| Protists (primary consumers) | Plant parasites | 1 |
| Protists (bacterivores) | Bacterivorous protists | 2 |
| Protists (secondary consumers) | Predatory and omnivorous protists | 2 |
| Collembola (omnivores) | Collembola | 2 |
| Oribatid mites (omnivores) | Oribatid mites | 2 |
| Other belowground arthropods (primary consumers) | Herbivorous, detritivorous and fungivorous Coleoptera and Hemiptera | 2 |
| Belowground arthropods (secondary consumers) | Omnivorous or carnivorous Hemiptera, Coleoptera and Araneae | 3 |

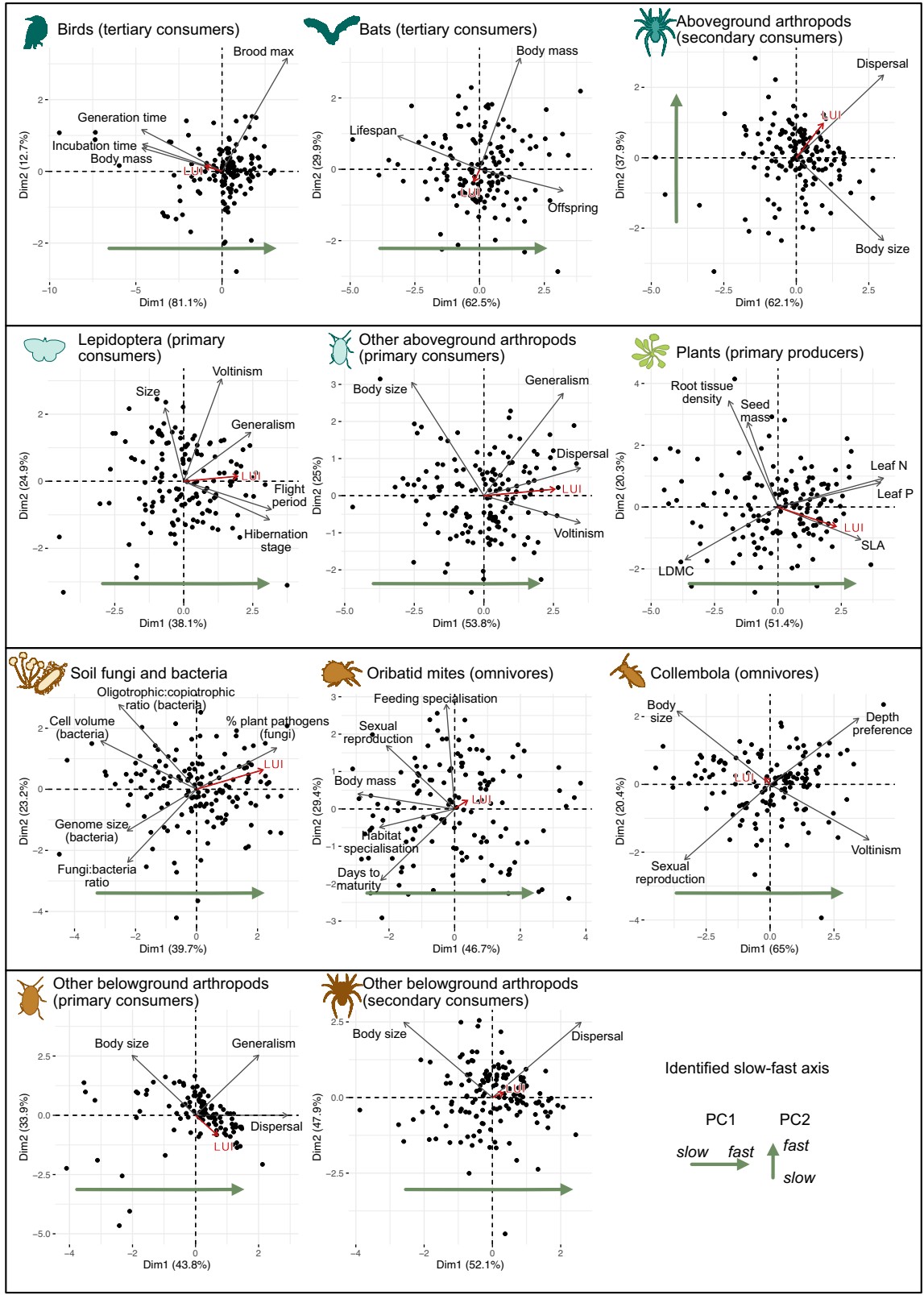

## Slow-fast trait continuum across trophic levels

Next, we explored the degree of correspondence between the community-level 'slow-fast' trait axes of individual guilds. The data supported our overarching hypothesis H1: most of the guild-level 'slow-fast' axes were closely correlated with each other and with the land-use intensity gradient. A Principal Components Analysis (PCA)

conducted on the slow-fast axes of all guilds with sufficient data availability (all except belowground arthropods (primary consumers), see Methods) revealed that 25% of the combined variation in all the guilds' slow-fast axes was explained by the first PC axis of variation. This axis was strongly correlated with land-use intensity (Pearson correlation $r = 0.74$, $p < 10^{-20}$, Fig. 3a). We also observed a significant

**Fig. 2 | Identification of slow-fast axes for each guild.** PCA were run on the CWM of traits hypothesised to be related to the slow-fast strategies in each guild to identify a common slow-fast axis (green arrows). This was the first PC axis for all guilds, except aboveground secondary consumer arthropods. The evidence supported the hypothesised slow-fast axis for most guilds, and partially supported it for three guilds. For Lepidoptera, there was no response from body size, contrary to expectations. Body size was measured as wing length, an indicator of overall body size (which is expected to be a "slow" trait; smaller size helps to survive disturbance) but also of dispersal ability (a "fast" trait, as larger wings promote recolonisation after disturbance). Both effects might cancel each other leading to no response of size. For secondary consumer, aboveground arthropods, dispersal and body size are slightly confounded

because larger body size increases dispersal abilities. However, these two traits are opposed on the second axis of the PCA, which we use as our slow-fast index. The last guild for which we found only partial support for the slow-fast axis was bats. High body mass is usually considered a 'slow' trait, and is expected to be positively correlated to lifespan and negatively to the number of offspring (trade-off between survival and reproduction). However, hibernation saves resources and leads, in hibernating bats (most of the species observed in our study), to a correlation between number of offspring and body mass[228], leading to the results observed here. LUI: land-use intensity. Icons were acquired and adapted from Phylopic.org (artists: M. Dahirel, B. Lang, M. Crook, J. A. Venter, H. H. T. Prins, D. A. Balfour, R. Slotow, T. M. Keesey, A. A. Farke, Y. Wong, G. Monger).

and positive correlation between land-use intensity and the slow-fast axis score of nine (out of 14) guilds ($p$ ranging from $1.10^{-14}$ to 0.04 and $|r| > 0.2$), with this correlation being particularly strong at low trophic levels. In contrast to our expectations, the highest trophic levels tended to have 'slower' traits at high land-use intensity (birds, $r = -0.2$, $p = 1.5\,10^{-2}$). We also tested for the existence of the whole community slow-fast axis by running a PCA on all considered traits from all guilds simultaneously (rather than aggregated into guild-level slow-fast axes, themselves extracted from PCAs). This analysis also supported the existence of a whole community slow-fast axis, with 'slow' and 'fast' traits clearly separated across sites (Fig. 3b).

The observed covariation in slow-fast traits across guilds demonstrates that land-use intensity drives differentiation between entire communities, with 'slow' and 'fast' communities characterised by distinct trait syndromes (Fig. 3, Table S2). This conclusion was supported by the analysis of individual traits' responses to land-use intensity, which shows that the CWMs of 54% of individual traits responded to land-use intensification in the hypothesised direction, and only two (4%) responding in an opposing direction (Fig. 4). 'Slow' communities are found at low resource availability and infrequent disturbance, and are characterised by slow-growing resource-conservative plants (correlation of leaf dry matter content with land-use intensity: $r = -0.28$, $p = 7\,10^{-4}$; root tissue density: $r = -0.22$, $p = 7\,10^{-3}$). Their soil microbial communities are dominated by fungi rather than bacteria (fungal:bacteria ratio: $r = -0.28$, $p = 3\,10^{-5}$), consistent with previous observations that low nutrient availability selects for fungal-dominated communities and slow-growing plants[21,22]. Bacterial communities in 'slow' communities also contain a higher proportion of small-celled organisms with large genomes (cell volume: $r = -0.30$, $p < 10^{-8}$; genome size: $r = -0.14$, $p = 2\,10^{-4}$), an adaptation to growth in nutrient-poor (oligotrophic) conditions[58] (oligotrophic:copiotrophic ratio: $r = -0.14$, $p = 0.02$). In contrast to the overall trend, birds had slower strategies in high intensity than low-intensity grasslands (slower generation length, $r = 0.19$, $p = 0.02$ and larger body size, $r = 0.23$, $p = 0.005$).

At the 'fast' end of the spectrum, high resource availability is associated with fast-growing plants with high nutrient content (leaf N content: $r = 0.46$, $p < 10^{-8}$, leaf P content: $r = 0.51$, $p < 10^{-8}$), which is often related to lower levels of physical and chemical defence[59]. This strategy was associated with a higher dominance of pathogenic fungi[60] (proportion of pathogenic soil fungi, $r = 0.37$, $p < 10^{-8}$) and plant pathogenic soil protists[61] ($r = 0.46$, $p < 10^{-8}$). In addition, high land-use intensity selected for smaller body size in four guilds (bacterivore protists: $r = -0.29$, $p = 3\,10^{-4}$; secondary consumer protists: $r = -0.39$, $p < 10^{-8}$, aboveground arthropods (primary consumer) size: $r = -0.24$, $p = 4\,10^{-3}$; belowground arthropods (primary consumers): $r = -0.24$, $p = 2\,10^{-3}$) and faster reproduction[19] in two guilds (voltinism of Lepidoptera (primary consumers): $r = 0.21$, $p = 0.01$; and other aboveground arthropods (primary consumers): $r = 0.56$, $p < 10^{-8}$). This response is likely driven by disturbance, which causes higher mortality in large-bodied arthropods[62,63], and favours early reproductive maturity and fast reproduction as organisms can reproduce before disturbance, and population

numbers can recover quickly afterwards. In addition, dispersal ability was greater in the fast communities of high land-use intensity (dispersal ability of aboveground arthropod (secondary consumers): $r = 0.33$, $p = 3\,10^{-5}$, aboveground arthropods (primary consumers): $r = 0.37$, $p < 10^{-8}$). This higher dispersal capacity enables faster recolonisation after disturbance[15,62].

The slow-fast trait responses of some consumer guilds was potentially driven not directly by resources and disturbance, but indirectly via losses in plant diversity associated with increased fertilisation[64,65]. This commonly leads to the loss of associated specialist plant species that are typically found in slow ecosystems[33,66,67]. As such specialist plant species often have higher degrees of physical and chemical defences, they tend to be associated to specialist herbivores[68]. In our study, this effect was manifested by an increased dominance of generalist species with land-use intensity, both among Lepidoptera and other aboveground arthropods (primary consumers) (feeding generalism of Lepidoptera: $r = 0.30$, $p = 3\,10^{-4}$; and of aboveground arthropods (primary consumers): $r = 0.34$, $p < 10^{-8}$).

Changes in community-level traits along environmental gradients can be due to changes in species identity (taxonomic turnover) or variation in species relative abundance. In our case, we found that both factors were responsible for the observed slow-fast responses to land-use intensity within the different guilds (average turnover across guilds: $0.95 \pm 0.03$; average nestedness: $0.03 \pm 0.03$, Table S3).

## Land-use intensity effect on community-level slow-fast response

The observed entire community-level slow-fast continuum could be driven by a common, but independent, response of individual guilds to land-use intensity, or mediated by cascading bottom-up trophic interactions between guilds, and we hypothesised that both pathways were important (Hypothesis 2). To test this, we used Structural Equation Modelling (SEM) to assess the effect of these pathways on the 'slow-fast' axes of the 13 guilds. We found that both pathways were important, supporting Hypothesis 2: slow-fast strategies axes are shaped both directly via shared responses to land-use intensity and indirectly via trophic interactions. Land-use intensity showed significant direct effects on the 'slow-fast' axis of eight guilds out of 13 (average estimate: $0.20 \pm 0.06$), and indirect, i.e., trophically-mediated, effects on six guilds ($0.07 \pm 0.02$). The pathways combined to make significant total effects of land-use intensity ($0.27 \pm 0.07$) on nine out of the 13 guilds (Fig. 5a).

Next, we obtained a general assessment of how land-use intensity affected the slow-fast axis of each trophic level, by averaging the direct, indirect and total effects for all guilds at each trophic level (producers vs. primary vs. secondary vs. tertiary consumers). This showed that the 'slow-fast' axes of lower trophic levels responded more strongly than higher trophic levels to land-use intensity, via both the direct and indirect paths. While there was a significant total effect of land-use intensity on all trophic levels, except belowground secondary consumers (Fig. 5b), the direct ($p = 2\,10^{-3}$), indirect ($p = 0.01$) and total ($p = 4\,10^{-4}$) effects all significantly decreased in intensity with trophic levels, indicating more synchronous trait responses at lower trophic levels (Fig. 5c).

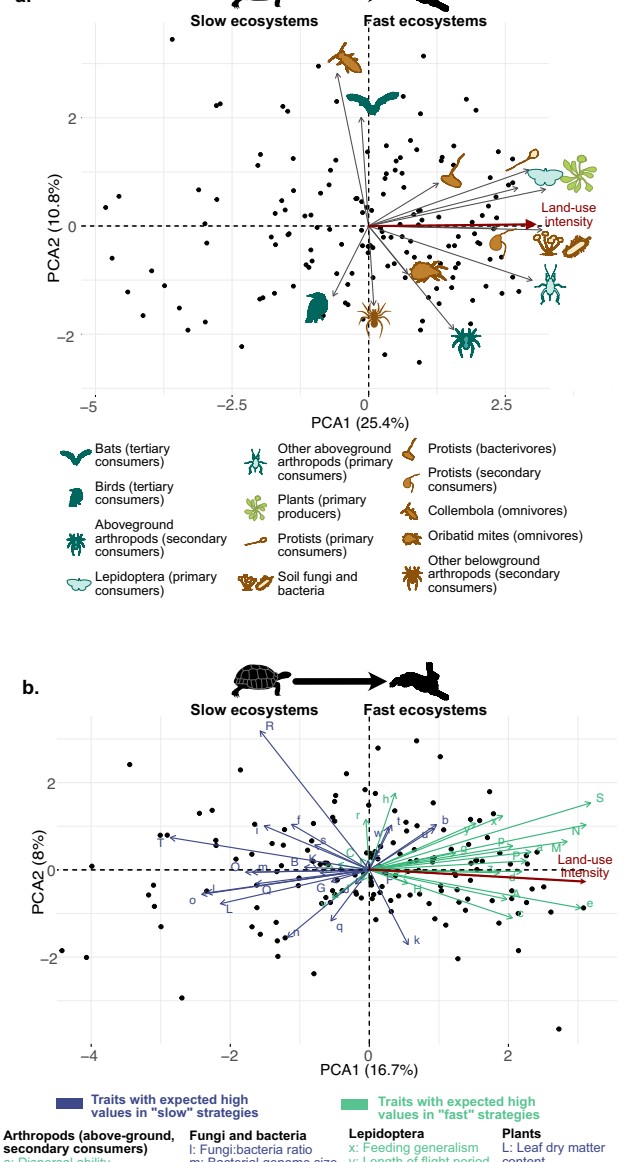

**Fig. 3 | The slow-fast trait axis of individual guilds is strongly related to land-use intensity. a** Whole community slow-fast axis, based on guild-level slow-fast axes. The variables included in the PCA are the slow-fast axes of each guild, which were estimated as the PC axis that best represents the hypothesised slow-fast axis (PC1 in 90% of cases) in a PCA of the selected CWM traits expected to be related to slow-fast strategies for each guild individually. These guild-level PCA can be found in Fig. 2. Land-use intensity (which is projected on the PC axes, shown in dark red) is strongly associated with Axis 1. Belowground guilds are shown in brown, aboveground guilds in blue, and plants in green. **b** Whole community slow-fast axis, based on all traits. In contrast to the results shown in **a**, the PCA was conducted on all traits, at the CWM level. Each trait was weighted as 1/n, with n with n the number of traits available for the guild, so that all guilds are weighted equally. Traits expected to be 'slow' are coded in blue, 'fast' in green. Sample size: 150 (sample sizes for each individual guild are shown in Fig. 4, missing values were imputed to run the PCA, see Methods). Icons were acquired and adapted from Phylopic.org (artists: M. Dahirel, B. Lang, M. Crook, J. A. Venter, H. H. T. Prins, D. A. Balfour, R. Slotow, T. M. Keesey, A. A. Farke, Y. Wong, G. Monger).

microbes), land-use intensity or taxonomic diversity, $R^2$ from 0.11 to 0.26, Table S4). The exception to this was fungal-bacterial ratio, which was correlated with the whole slow-fast axis ($r = -0.84$, $P < 10^{-8}$) and better explained the functions slow-fast axis ($R^2 = 0.47$, Figs. S5 and S6). This was likely due to the prevalence of soil-related measures in our selected set of functions. Additionally, some individual functions were more strongly associated with specific guilds or traits (e.g., fast plant community traits were positively associated with biomass production (Fig. S5)). Results were similar when using the all traits slow-fast axis (PC1 in Fig. 3b) as the main community slow-fast axis (Fig. S4).

A mediation analysis found that the effect of land-use intensity on the ecosystem functioning 'slow-fast' axis was both direct and mediated by the functional traits 'slow-fast' axis, as both direct and indirect paths were significant (Fig. 6b). Together these results support Hypothesis 3 and suggest that the whole community fast-slow trait continuum is linked to a slow-fast axis of whole ecosystem functioning.

### Sensitivity analyses

While there were strong hypothetical reasons to expect body size to be a key trait driving fast-slow variation at the community level, it can respond to a range of drivers and drive life-history variation in the absence of other trait responses; as a result, the effect of body mass is often removed in analyses that seek to identify life-history trade-offs[69,70]. To assess the robustness of our results to this trait, we conducted additional analyses in which we excluded all body size data (resulting in the exclusion of bacterivorous and predatory protists as other trait data were not available for these groups). Even in the absence of body size were still able to identify a strong guild-level slow-fast axis for most guilds, except Oribatid mites. This resulted in somewhat weaker, but still consistent results regarding the synchrony of slow-fast axes across guilds and the effect of the whole community slow-fast axis on ecosystem functioning (Tables S8–S10, Figs. S7–S9).

Other sensitivity analyses included checking the impact of using raw CWM data (uncorrected for environmental covariates) and unweighted (instead of using abundance-weighted) trait values. These are discussed in the supplementary analyses (Tables S11–S18; Figs. S10–S20); they show overall weaker, but consistent results with those presented here.

## Discussion

Our results provide strong evidence for the existence of synchronous, whole ecosystem-level responses to environmental drivers, and more specifically the existence of a slow-fast axis of variation at the level of entire multitrophic communities. We show that there are similar functional responses to resource availability and disturbance across taxa and trophic levels, from large organisms with slow reproduction and slow dispersal at low land-use intensity to small, fast-paced organisms in more intensively managed sites. The findings of this axis

### Slow-fast ecosystem functioning response

To test our third hypothesis, that the community-level slow-fast continuum was related to whole ecosystem function, we first conducted a PCA based on 15 ecosystem functions related to carbon fluxes, nitrogen fluxes, biomass production and decomposition. This showed that ecosystem functions covary, from 'slow' sites characterised by slow decomposition, biomass production and low enzyme activities to 'fast' sites with faster nutrient cycling (first axis of the PCA: 29% variance, Fig. 6a). This whole ecosystem functioning slow-fast axis was better explained by the entire community 'slow-fast' traits axis ($r = 0.4$, $R^2 = 0.34$, $p < 10^{-8}$) than by other hypothesised drivers of ecosystem functioning (single guild community traits measures (plants and

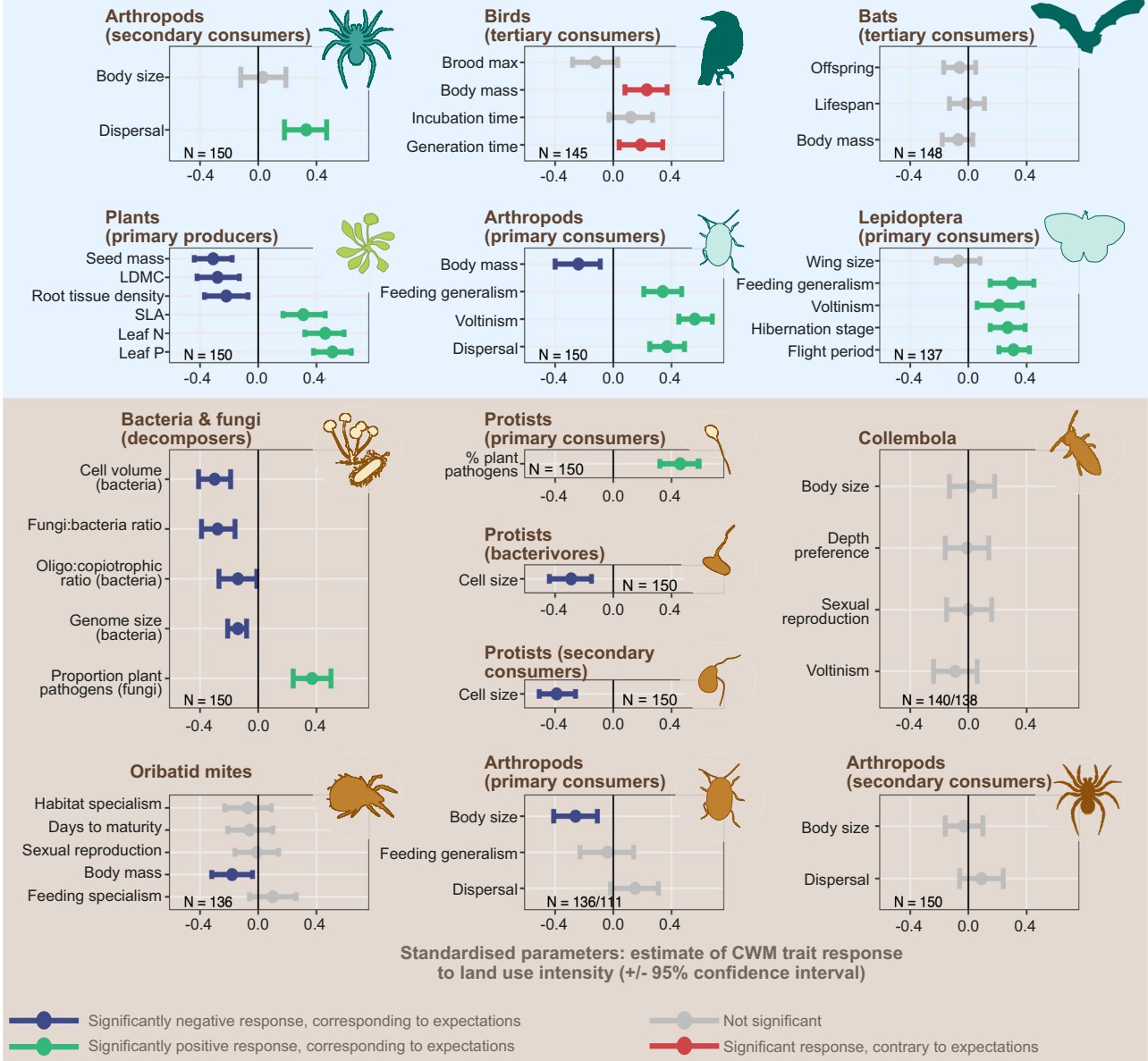

**Fig. 4 | Observed responses of individual traits at the guild-level (CWM) to land-use intensity, shown as estimated parameter +/− 95% confidence intervals.** The responses shown were extracted for each trait individually as the slope ± confidence interval of a linear model with the CWM trait as a response to land-use intensity after correction for other environmental covariates (see Figure S1). *P*-values (two-sided *t-tests*) were adjusted for multiple testing. Land use intensity explained between 0 (non-significant traits) and 40% (Aboveground primary consumer arthropods: voltinism) of the total variation in individual trait CWM (Table S2). Belowground guilds are shown over a brown background, aboveground guilds over blue. A standardised response below 0 (blue lines) indicate a negative response to land-use intensity, in accordance with hypotheses (expected higher trait values in 'slow' communities). Responses above 0 (green line) indicate a positive response to land-use intensity, in accordance to hypotheses (expected higher trait values in 'fast' communities). Red lines indicate traits with opposite responses to that hypothesised. 95% intervals crossing the 0 line indicate no significant response (grey lines). The number of replicates (sites with available data) are shown for each guild. Collembola: 138 for voltinism, 140 for the other traits. Arthropods belowground (primary consumers): 111 for feeding generalism, 136 for other traits. Icons were acquired and adapted from Phylopic.org (artists: M. Dahirel, B. Lang, M. Crook, J. A. Venter, H. H. T. Prins, D. A. Balfour, R. Slotow, T. M. Keesey, A. A. Farke, Y. Wong, G. Monger).

extend earlier studies which investigated the coordinated responses of a few trophic guilds[24–26,71] and demonstrate a previously unrecognised emergent property of multitrophic community assembly. We also demonstrate that entire community-level synchrony is driven by both the shared responses of individual guilds to land management and trophically-mediated cascades. Synchrony was stronger at the lower trophic levels which make up the vast majority (>99%) of community biomass[72], with weaker effects for the less abundant organisms of higher trophic levels (e.g., birds). This is likely due to higher trophic level organisms, e.g., birds and bats, being less dependent on local

management conditions and instead responding to larger-scale drivers, such as landscape composition[33]. Belowground organisms also tended to show weaker responses than aboveground ones, likely due lower trait data availability weakening our capacity to detect a response and lower disturbance of the soil environment than the aboveground environment by the intensification studied here - mowing and grazing[33]. Finally, we provide evidence that this whole community strategy variation mediates the effect of land-use intensity on overall ecosystem functioning. More specifically, communities dominated by faster, non-conservative strategies (e.g., faster reproduction,

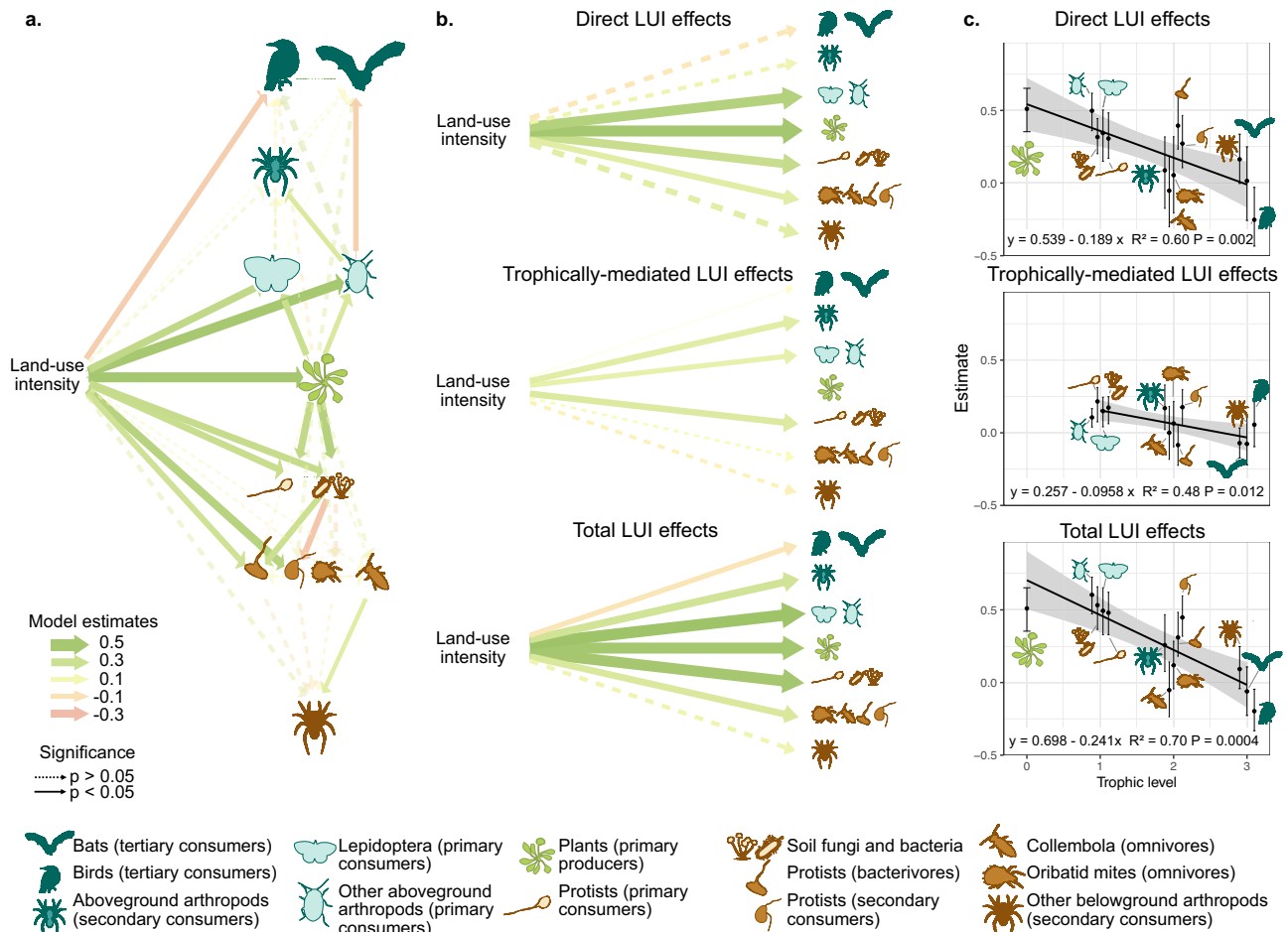

**Fig. 5 | Direct and trophically-mediated effects of land-use intensity on the slow-fast axis of individual trophic guilds. a** Full SEMs including all guilds. Two independent models were fitted for below- and aboveground guilds; plants were included in both. **b** Average direct, trophically-mediated (indirect) and total land-use intensity (LUI) effects on each trophic level (averaged across guilds within each trophic level from the SEM estimates shown in **a**. **c** Decreasing direct, indirect and total land-use intensity effects with trophic level. Each dot represents the estimated effect (mean ± standard error) of land-use intensity on the slow-fast axis of each individual guild in the full SEM. Sampling size per guild: 150 for all guilds except Oribatid mites (136), Collembola (138), Lepidoptera (137), birds (145) and bats (148). In **a** and **b** colours indicate the path estimate value from negative (orange) to positive (green); *p*-values are extracted from two-sided *z*-tests; in c two-sided *t*-tests; they were not corrected for multiple testing. Precise *p*-values can be found in Tables S6 and S7. Belowground, primary consumer arthropods were excluded (see Methods section). Icons were acquired and adapted from Phylopic.org (artists: M. Dahirel, B. Lang, M. Crook, J. A. Venter, H. H. T. Prins, D. A. Balfour, R. Slotow, T. M. Keesey, A. A. Farke, Y. Wong, G. Monger).

---

shorter lifespan) have faster metabolic rates per unit biomass, a higher digestibility and/or tissue resource concentrations and rapid turnover of tissues. All these factors lead to a faster transfer of resources from organic to inorganic pools and between trophic levels[28,73,74]. In terms of ecosystem functioning, this equates to greater productivity and gas fluxes, higher soil enzyme activities, and faster decomposition and rates of organic matter mineralisation. This extends previous findings that demonstrate the linkages between traits of individual guilds and ecosystem functions[23,36,37].

The theory describing slow and fast strategies, and the identity of their defining traits, is very well developed in some organisms such as plants[8,30,75]. In most other taxa, however, such theories are only emerging (e.g. microorganisms[13], arthropods[76]). Here, we built upon both in-depth expert discussions and observational studies to define and test hypotheses on the responses of individual taxa to resource availability and disturbance. While this allowed us to generate insights into possible slow-fast strategies for multiple functional groups our need to do this also highlights the need for further theory development on the functional strategies of many taxa. Such theoretical advances would provide key building blocks towards the improved

understanding of the response of slow-fast functional strategies to environmental drivers at the level of individual organisms, guilds and entire communities. Such theory could also support the integration of other axes of functional variation, which have been described for individual taxa at species[3,31], and community level[77].

Our results highlight that the average functional strategy of guilds, as represented by the CWM of multiple traits, can be meaningfully related to land-use intensity, linkages with other guilds, and measures of ecosystem functioning. By focusing on CWM trait means, we did not consider the role of strategy variation within a community, which can be considerable[78]. This functional diversity can represent multiple winning strategies within a community and/or niche differentiation from the optimal strategy, and thus the avoidance of competitive exclusion. Functional and taxonomic diversity both within and across taxa has been shown to play an important role in driving multitrophic interactions[79] as well as ecosystem functioning[80] and plant diversity is commonly related to higher levels of ecosystem functioning[81–83]. Previous studies have explored the relative response of dominant strategies, versus their variability, in response to both environmental factors[84] and as drivers of ecosystem functioning[85].

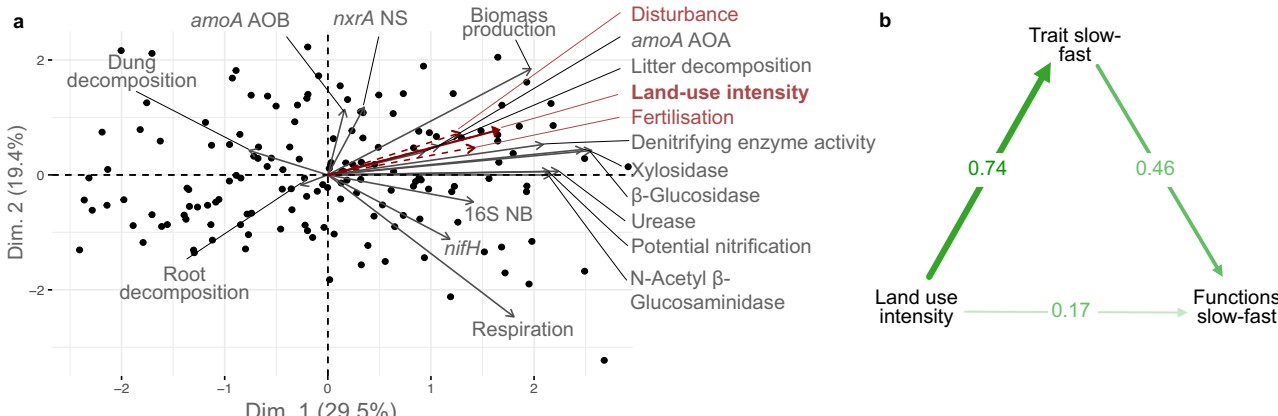

**Fig. 6 | The ecosystem functions slow-fast axis is linked more strongly to the trait slow-fast axis than to land-use intensity. a** PCA of the 15 selected ecosystem functions. Functions within each bundle (carbon fluxes, nitrogen fluxes, biomass production and decomposition) are weighted inversely to the number of functions in the bundle. The first axis shows a slow-fast axis from slow decomposition, biomass production and low enzyme activities to faster nutrient cycling. Land-use intensity was projected onto the PCA, and is strongly associated with axis 1. **b** Direct and indirect links between land-use intensity, the entire community traits slow-fast axis (identified in Fig. 3a: PC1 of the PCA) and the ecosystem function slow-fast axis.

DEA denitrification enzyme activity; urease, β glucosidase, N-Acetyl β glucosaminidase, xylosidase: activity of the respective enzyme; *amoA*_AOA: abundance of ammonia oxidation gene of archaea; *amoA*_AOB: abundance of ammonia oxidation gene of bacteria; *nifH*: abundance of nitrogen fixation gene in soil bacteria; *nxrA* NS: abundance of nitrite oxidation gene of Nitrobacter bacteria; 16S NB: abundance of nitrite-oxidizing bacteria; dung decomposition; plant aboveground biomass production; litter decomposition; root decomposition. Sample size: 150 sites (sample sizes for each individual guild are shown in Fig. 4, missing values were imputed to run the PCA, see Methods section).

Expanding this approach from single groups to whole communities could provide new insights into the behaviour of functional diversity as a multitrophic property that may drive ecosystem structure and functioning at a 'whole systems' level.

It is worth noting that the whole ecosystem 'slow-fast' gradient observed here was manifested across a relatively short environmental gradient - all sites were temperate agricultural grasslands. It remains to be seen whether the results hold across time, as suggested by dynamic linkages between plant and microbial traits in time[22], and for other systems, in particular a diversity of longer, natural and more orthogonal gradients of disturbance and resource availability. We hypothesise that across stronger environmental gradients of climate and soils, e.g., between biomes, and when incorporating a more comprehensive array of slow-fast traits, ecosystem-level trait synchrony will be even more marked. Correspondence between global gradients in climate and soils and plant community-level trait measures[86] support this idea, especially given that the traits of other trophic levels are often strongly associated with plant communities[22–24]. The close relationship between our ecosystem slow-fast axis and fungal-bacterial ratio also suggests that this axis is compatible with earlier literature pointing to a slow-fast functioning gradient from fungal-to bacteria-dominated communities[21,28,87] and that fungal-bacterial ratio could act as an indicator variable for the whole ecosystem slow-fast continuum. Such synchrony may also correspond to recently described global trends in the covariance of multiple ecosystem functions[39,88]. It could also be related to other widely reported patterns of ecological change, such as the global "community downsizing" of animal communities that occurs in response to both climate change and anthropogenic disturbance[89,90] and its impact on ecosystem functioning[91], though further theoretical integration would also be required. When combined with these results, our findings highlight the potential for a new generation of whole ecosystem-level studies that go beyond the description of single trophic guild-level strategies to characterise communities, or single functions to characterise ecosystem responses. Instead, these would work at the level of universal ecosystem-level functional types and axes[38,39]. This may form the basis of a new typology for classifying ecosystems, and provide a generalisable, predictive and simple means of describing their responses to land-use and environmental change.

## Methods
### Study area
The study was conducted as part of the long-term Biodiversity Exploratories project (www.biodiversity-exploratories.de). Data was collected in 150 grassland plots in three regions of Germany: the Schwäbische Alb plateau and UNESCO Biosphere Reserve in southwestern Germany; the Hainich National Park and surrounding areas in central Germany (both are hilly regions with calcareous bedrock) and the UNESCO Biosphere Reserve Schorfheide-Chorin in the post-glacial lowlands of north-eastern Germany. The three regions differ in climate, geology and topography, but each is characterised by a gradient of grassland land-use intensity that is typical for large parts of temperate Europe[49]. In each region, 50 plots (50 m × 50 m) were chosen in secondary wet, mesic and dry grasslands by stratified random sampling from a total of 500 candidate plots on which initial vegetation, soil and land-use surveys were conducted. This ensured that plots covered the whole range of land-use intensities and management types, while minimising confounding factors such as spatial position or soil type.

### Land-use intensity
Land-use intensity was assessed annually via questionnaires sent to land managers in which they reported the level of fertilisation (kg N ha$^{-1}$ yr$^{-1}$), the number of mowing events per year (from one to three cuts, starting in May and continuing up to September/October in the most intensive plots), and the number and type of livestock and their duration of grazing (number of livestock units ha$^{-1}$ yr$^{-1}$)[49,51]. Mowing and grazing intensities determine the frequency, and intensity, at which aboveground biomass is removed; and thus represent the intensity of disturbance in the plot. Fertilisation provides additional nutrients, and is typically mostly applied in naturally productive plots: it thus represents a resource availability gradient. In our study system, mowing and fertilisation intensities are positively correlated ($r = 0.7$), while grazing and mowing intensities are negatively correlated ($r = -0.6$). Thus, independent effects of each land-use component, and the respective effect of resource availability and disturbance, cannot be reliably estimated. We therefore used a compound index of land-use intensity, characterising a combined resource availability and disturbance gradient. The land-use intensity index (LUI) was calculated as the square-root-transformed sum of standardised measures of

global mowing, fertilisation and grazing intensities across the three regions for each year[92]. We calculated the mean LUI for each plot over the years 2008–2018 because this reflects the average LUI around the years when most of the data was collected. At low LUI (0.5–0.7), grasslands are typically neither fertilised or mown, but grazed by one cow (>2 years old) per hectare for 30 days (or one sheep per hectare for the whole year). At an intermediate LUI (around 1.5), grasslands are usually fertilised with less than 30 kg N ha$^{-1}$y$^{-1}$, and are either mown twice a year or grazed by one cow per hectare for most of the year (300 days). At a high LUI (3), grasslands are typically intensively fertilised (60–120 kg N ha$^{-1}$y$^{-1}$), are mown 2–3 times a year or grazed by three cows per hectare for most of the year (300 days), or are managed by a combination of grazing and mowing. The study area did not cover very high intensity grasslands (e.g., cut more than three times per year and ploughed annually).

In some figures, LUI was added as a supplementary variable to e.g., PCAs, along with a 'fertilisation' variable and a 'disturbance' index. The fertilisation variable is the standardised fertilisation value. The disturbance index is calculated as the square-root of the sum of mowing and grazing intensities, both standardised by dividing by their overall average.

### Acquisition of environmental covariates

To measure pH and soil texture, composite samples were taken in 2011 in all 150 plots, by mixing 14 mineral topsoil samples (0–10 cm, using a split tube manual soil corer with 5 cm diameter). Ten g of sieved and air-dried soil were mixed with 25 ml 0.01 M CaCl$_2$ solution and shook for 2 h. Afterwards the pH of the soil suspension was measured using a glass electrode. The pH of each sample was measured twice. We determined soil texture by separating soil particles into sand (2–0.063 mm), silt (0.063–0.002 mm) and clay (<0.002 mm) by sieving and sedimentation (DIN-ISO 11277). Open datasets from Biodiversity Exploratories: refs. 93–95.

The topographic wetness index (TWI) combines measures of upslope contributing area (determining the amount of water received from upslope areas) and slope (determining the loss of water from the site to downslope areas) and has been shown in previous analyses to be a better predictor than local humidity measures[33]. It is defined as ln(a/tanB), where a is the specific catchment area (cumulative upslope area which drains through a Digital Elevation Model (DEM), http://www.bkg.bund.de) cell, divided by per unit contour length) and tanB is the slope gradient in radians calculated over a local region surrounding the cell of interest. TWI was calculated from raster DEM data with a cell size of 25 m for all plots, using GIS tools (flow direction and flow accumulation tools of the hydrology toolset and raster calculator). The TWI measure used was the average value for a 4 × 4 window centred on the plot, i.e., 16 DEM cells corresponding to an area of 100 m × 100 m. Dataset from the Biodiversity Exploratories: ref. 96.

Finally, we measured mean annual temperature as the temperature 2 m aboveground level in each plot, aggregated at the year level and averaged between 2008 and 2015. Open dataset from Biodiversity Exploratories: ref. 97.

### Sampling protocols by trophic guild

In each plot, we measured the relative abundance of multiple guilds using standard methodology. Most data was extracted from an aggregated diversity dataset from the Biodiversity Exploratories[98,99]. Original datasets are listed below.

**Vascular plants.** We sampled vascular plants in an area of 4 m × 4 m in all 150 plots, and estimated the percentage cover of each occurring species every year from 2008 to 2019. Open dataset from Biodiversity Exploratories: ref. 100.

**Soil fungi and bacteria.** Composite soil samples were taken in 2011, 2014 and 2017 in all plots, by mixing 14 mineral topsoil samples (0–10 cm, using a split tube manual soil corer with 5 cm diameter).

For bacteria the analysis only included data from 2011 (148 plots) and 2014 (150 plots). 10 g of the homogenised soil was put immediately on liquid nitrogen and stored until RNA extraction. RNA was extracted using a custom protocol (Lueders protocol). Total RNA was isolated from soils and reverse transcribed into cDNA. Amplicons of the V3 region of the 16S rRNA gene were sequenced on an Illumina Hiseq platform using universal bacterial primers.

For fungi, total microbial DNA was isolated from the bulk soil sample using a MoBioPowerSoil DNA Isolation Kit. A PCR approach was used to amplify fungal ITS-rDNA by using the primer pair ITS4/fITS7, containing the Illumina adapter sequences. PCR products were then purified, cleaned and sequenced using Illumina MiSeq. Data was available for 150 plots in 2011, 2014 and 2017.

Open datasets from Biodiversity Exploratories: refs. 101–105. Other dataset from the Exploratories: ref. 106.

**Soil protists.** Soil DNA was extracted in 2011 and in 2017, from 400 mg of soil (not sieved), 3- to 6-times, using the DNeasy PowerSoil Kit (Qiagen GmbH, Hilden, Germany) following the manufacturer's protocol. Data for one plot in 2011 was missing, data for all 150 plots was available in 2017. Specific primers for Cercozoa and Endomyxa were used to amplify, by two semi-nested PCRs, the V4 region of the 18 S rRNA gene. Libraries were prepared using TruSeqDNA PCR-Free (Illumina, San Diego, CA, US). Sequencing was performed with a MiSeq v3 Reagent kit of 300 cycles (on a MiSeq Desktop Sequencer, Illumina). The bioinformatics pipeline was conducted with Mothur v. 39.5. After assembling and quality filtering, sequences were clustered at 97% similarity, and rare clusters (<0.01% of the reads) were removed. OTUs were identified in the PR2 database using BLASTn with an e-value of 1$^{e50}$ and keeping only the best hit. Chimeric OTUs were identified using UCHIME and removed[61].

Open datasets from Biodiversity Exploratories: refs. 107,108.

**Collembola and Oribatid mites.** In 2019, we sampled four soil cores of 4.5 cm × 10 cm per plot. Collembola and Oribatid mites were then extracted following a Kempson extraction and identified. Collembola data was available for 140 plots. For Oribatid mites, it was available in 149 plots but some plots had to be excluded due to low trait data coverage (see below), resulting in 136 used plots in total. Datasets from the Biodiversity Exploratories: refs. 109,110.

**Lepidoptera.** Lepidoptera were recorded using sweep netting along transects of 300 m during 30 min, done three times per plot in 2008. Data was available for 137 plots. Open dataset from the Exploratories: ref. 19.

**Other arthropods.** All arthropods of the herb layer were sampled twice per year between 2008 and 2017 in June and August to represent different phenological windows within the peak season of adult arthropod activity. Total number of plots sampled each year varied from 143 to 150, all plots were sampled at least eight years (average: 9.8 years). Based on monthly samplings at the beginning of the study, we identified these two months to represent the best trade-off between reducing sampling effort and covering most species. Arthropods were sampled by sweep netting along a 150-m long transect comprising three of the virtual borders of a site by conducting 60 double sweeps per site. Sweep netting was only conducted on days without rain, low wind speed and after morning dew had dried. All samples were sorted to order level in the laboratory. For taxonomic groups that occurred in larger numbers and for which expert taxonomists were available, adult specimens were identified at species level: Araneae, Coleoptera, Hemiptera

(Heteroptera and Auchenorrhyncha), Orthoptera. Species that spend a significant part of their lifecycle in the soil or litter were classified as belowground species. Open data from the Exploratories: ref. 111.

**Birds.** Birds were recorded using audio-visual point counts during breeding times (March-June), at the centre of each respective grassland plot (50 m × 50 m). This was done every year between 2008 and 2012 and again in 2018. Records were available for 72–128 plots per year, with four years of records available per plot on average. Open datasets from the Biodiversity Exploratories: refs. 112–117.

**Bats.** Acoustic recordings were conducted along the edges of each plot, using a combination of point-stop and transect monitoring in 2009 and 2010. Point-stops were located at each corner of the plot. Transects were walked slowly and in a direct line between the point stops. Survey time at point-stops and during transect walks were 3 min each. This resulted in a total survey time of 24 min along a 200 m transect per plot. Recordings were made using a Pettersson D 1000× ultrasound detector (Pettersson Electronic AG, Uppsala, Sweden). Echolocation calls were identified to species level or to Sonotype using the software Avisoft SAS Lab Pro, Version 5.0.24 and onward (Raimund Specht, Avisoft Bioacoustics, Berlin Germany). Records were available for 148 plots in total, including 134 in 2009 and 130 in 2010. Open datasets from Biodiversity Exploratories: refs. 118,119.

## Aggregation to functional guilds

These taxa were compiled into functional guilds in which trophic status was broadly comparable (e.g., aboveground primary consumers or belowground omnivores). Within each functional guild, organisms of different taxa were aggregated if trait data was comparable, and if not, the functional guild was classified at the taxonomic level (Table 1). Aboveground guilds included vascular plants (primary producers); Lepidoptera (primary consumers – incl. butterflies and day-flying moths); other aboveground primary consumers arthropods (incl. primary consumers of Orthoptera, Coleoptera, and Hemiptera); aboveground secondary consumers arthropods (including omnivorous Hemiptera, carnivorous Coleoptera, Araneae, Orthoptera and Hemiptera); bats (tertiary consumers); and birds (tertiary consumers, i.e., excluding herbivorous and granivorous birds). Belowground guilds included: bacterial and fungal communities (symbionts, decomposers and parasites); plant parasite protists (primary consumers); bacterivorous protists; secondary consumers protists (incl. predators and omnivores); Collembola (omnivores); Oribatid mites (omnivores); other herbivorous, detritivorous and fungivorous belowground arthropods (primary consumers, incl. Coleoptera and Hemiptera); and predatory and omnivorous, soil-dwelling arthropods (incl. Araneae, Coleoptera, and Hemiptera).

## Functional trait data acquisition and treatment

Although it is general in overall concept, and has been discussed in a wide range of contexts, the spectrum of slow-fast strategies and their associated trade-offs has a well developed, trait-specific theory for only a few specific taxa, such as plants. Such theories are also emerging in the literature for other taxa[13,120] but in many cases are not fully developed yet. Selecting appropriate traits thus required us to combine hypotheses from existing group-specific theories[8], more general ecological theories (e.g., r/K[1,4]), empirical observations, and expert knowledge. We conducted multiple expert consultation discussions within subteams of the authors to identify traits potentially related to a resource availability and disturbance, and thus potentially falling on a slow-fast axis, i.e., size, dispersal abilities, feeding niche, etc. Detailed hypotheses guiding the choice of traits for each guild can be found in Table S2.

Trait data was obtained from multiple sources. Abundance data and trait databases were matched using the GBIF taxonomy (packages

taxize, traitdataform[121]) when necessary. Guild-specific details on trait data acquisition and treatment are described below.

**Plants.** Aboveground plant traits were compiled from the TRY database[5,23,36,45,122–220]. TRY data was first cleaned to remove duplicates, non-mature, non-healthy plants and plants which were grown in non-standard exposition (e.g. shade) as well as non-standard trait measurements. The trait data was then subsetted by data from central Europe to avoid geographic bias. Finally, resulting trait values were averaged by contributing sources, with outlier sources being excluded; and were then averaged for each species[221]. Belowground trait data was obtained from pot experiments[122]. Where appropriate, data from synonyms was used. When individuals were identified at the genus level, average trait values for all other species from the considered genus found during the survey was used. Community-weighted means were calculated by excluding tree saplings which are not part of the stable grassland communities and do not have the (adult) traits usually reported in databases.

**Birds.** Bird trait data for body length and incubation time was obtained from the literature and own datasets[47,222–226]. Data on functional groups and trophic levels was extracted from ref. 225 and the Avonet dataset[47]. Identification of ground-nesting species followed ref. 223. Maximum longevity and generation length were extracted from ref. 222. While trait data was also available for herbivorous species, the abundance data was not sufficient to include this group as 46 plots did not have any primary consumers (primarily herbivorous, frugivorous, and granivorous species). Thus, only secondary consumers (including carnivores, invertivores, and omnivores whose diet includes vertebrates or insects) were included in the study.

**Bats.** Bat traits data was obtained from ref. 227 (morphological traits) and ref. 228 (life-history traits). Both datasets were then restricted to species occurring in Germany based on ref. 229.

Abundance data was based on acoustic recording, which did not always allow for species-level identification: some individuals were classified as genera (*Myotis sp.*, *Plecotus sp.*) or similarly acoustic groups (Nyctaloids). In that case, traits were attributed based on the average values of species from the same genus present in Germany, or on all species in the Nyctaloid group (*Nyctalus octule, Vespertilio murinus, Eptesicus serotinus, Nyctalus leisleri, Eptesicus nilssonii*), respectively. The interpolation was usually relatively conservative (trait mean > 2×trait sd).

**Arthropods: Araneae, Coleoptera, Hemiptera and Orthoptera.** Body size (in mm), feeding guild, stratum use and dispersal ability of Araneae, Coleoptera, Hemiptera and Orthoptera were assembled from various literature sources and validated in correspondence with taxonomic experts for the respective groups. These were augmented by published trait data on feeding generalism from ref. 52 and voltinism from multiple sources[3,230–232]. Species were classified as secondary consumers if one of their life stages was mostly carnivorous, and as primary consumers otherwise (i.e., all life stages primarily herbivorous, fungivorous or detritivorous). Species were classified as belowground if at least one of their life stages was primarily soil- or ground-dwelling, and aboveground otherwise. Voltinism was coded as a numeric variable (0.5: semivoltine, 1: univoltine; 1.5: uni- or bivoltine; 2: bivoltine; 3: multivoltine). Dispersal was coded as a numerical variable (in five gradations (from low to high: 0, 0.25, 0.5, 0.75, 1) based on flight ability and wing dimorphism for insects and ballooning for spiders). Body size was log-transformed to avoid non-normal distribution. When sampled individuals were only identified at genus level in the abundance dataset, they were attributed the average traits of the other species in the genus only when the trait data was homogeneous (i.e., dominant category if it is representative of 90% of the species in the genus

among the sample species, and if the mean is > 2×standard deviation for quantitative variables); and was considered as a missing value otherwise. Open dataset from the Biodiversity Exploratories: ref. 233.

**Arthropods: Lepidoptera.** Traits for Lepidoptera were compiled from multiple databases. We used as primary sources ref. 55 for moth traits and ref. 19 for butterfly traits. In addition, we completed this data with traits from ref. 54, and the European and Maghreb Butterfly Trait database[56]. When datasets disagreed, we primarily kept the data from ref. 55 which was specifically curated for German moths. Voltinism was coded as a numerical variable: 0.5 for semivoltine species, 1 for strictly univoltine species, 1.25 for univoltine species with partial generation, 1.5 for uni- or multivoltine species, and 2 for multivoltine species. Feeding generalism was coded as a numerical variable: 1 for monophagous species, 2 for oligophagous species (within one plant genus), 3 for oligophagous species (within one plant family), and 4 for polyphagous species. Body size was provided as different metrics for the different databases: estimated dry mass, forewing minimum and maximum length in ref. 54, forewing average length in ref. 56, and body mass and wing length in ref. 55. The most complete was the forewing maximum[54]; we used data imputation (function mice, using default parameters) to impute the missing values in this variable, using all other size-related variables as predictors. The distribution of the data was similar before and after imputation, indicating reliable imputation. Body size was then log-transformed. Overwintering stage was coded as a numeric variable, from 1 (overwintering as egg), 2 (larvae), 3 (pupa) and 4 (adult). When datasets disagreed, we retained the latest stage of development as overwintering stage. Flight period was measured as the maximum number of flight months for adults. Again, ref. 54 was the most complete database. We used this data when available, and imputed missing values (function mice, default parameters) based on additional data from ref. 56. Flight period was then log-transformed. We completed the resulting data for three additional species (*Lythria purpuraria*, *Zygaena carniolica*, *Aphantopus hyperantus*) based on https://lepiforum.org/https://lepiforum.org/ and http://www.pyrgus.de/. Open datasets from Biodiversity Exploratories: refs. 53,234,235.

**Arthropods: Collembola and Oribatid mites.** Data on adult body mass and habitat specificity was collected on specimens collected during sampling (see above). Habitat specificity was coded numerically from 1 (non-specific habitat, most generalist), 2 (soil-dwelling), 3 (surface-dwelling) and 4 (litter-dwelling, which are the most specific species). Feeding specialisation was coded numerically from 1 (omnivorous, most generalists), 2 (herbifungivorous), 3 (herbivorous) and 4 (fungivorous, most specialist). (V. Wolters and A Zaytsev, pers. comm). Time to maturity (in days) and reproduction type (sexual or parthogenetic) were provided by experts and completed from the literature and coded as binary variables. Seven plots had low trait data coverage (<50%); they were given NA values for all CWM values and not used in the analysis. Dataset from the Biodiversity Exploratories[236].

**Bacteria and fungi.** We used a published dataset[46] to characterise bacterial communities.

For bacteria, trait data was used at genus level when available. If no genus-level data was available, it was extrapolated to the order level if the values of all genera within the order were consistent (mean > standard deviation for quantitative traits, and >60% of all genera sharing one trait value for qualitative traits). If the data was not consistent within the order, it was kept as missing values. Then, cell volume was estimated as (cell radius)^23.14×cell length and log-transformed before further analysis. Other traits such as doubling time were considered but had very low coverage (e.g., for doubling time median of OTUs with available data over all plots <20%) and were excluded from the analysis. Community-weighted means were

calculated using OTUs numbers as an approximation for relative abundances. We completed these traits with community levels characteristics. For bacteria, we quantified the relative abundance (approximated as OTU number) of groups commonly classified as oligotrophs (Acidobacteria, Verrucomicrobia, Planctomycetes) to copiotrophs (Actinobacteria, Betaproteobacteria, Gammaproteobacteria, Proteobacteria, Bacteroidetes).

For fungi, we calculated the relative abundance of pathotroph fungi among all fungi. Finally, we also measured the PLFA-based fungal-bacteria ratio from 2011 and 2014 (150 plots each year); data that were first published by ref 22. This was done by sampling two g of field moist soil (from the same composite samples as described above) for lipid extraction and fractionation following the alkaline methylation method described in ref. 237. Samples were measured by gas chromatography (AutoSystem XL. PerkinElmer Inc., Massachusetts, USA) using a flame ionization detector, an HP-5 capillary column and helium as the carrier gas. Fatty acid nomenclature used was described by ref. 238. Total bacterial PLFAs were calculated as the sum of Gram-positive (a15:0, i15:0, i16:0 and i17:0) and Gram-negative (cy17:0 and cy19:0)[239] plus the FAME 16:1ω?7 which is widespread in bacteria in general. Fungal biomass was represented by the PLFA 18:2ω?6.,9[239]. Open datasets from the Biodiversity Exploratories: refs. 104,105.

**Protists.** Genus-level protist trait data was obtained from ref. 240 and completed for size class data by K. Dumack. Size classes were coded numerically: 1 (<10 microns), 2 (11-30 microns), 3 (31–50 microns), 4 (>51 microns). Trophic levels were classified as plant parasites, bacterivores, or secondary consumers (non-plant parasites, eukaryvores, omnivores).

Primary consumers (plant parasites) had only one trait combination (size class 1) so we characterised these communities only by the relative abundance of plant parasites among all protists.

## Measurement of ecosystem functions
We selected 15 ecosystem function indicators from five bundles of related functions: decomposition (dung, litter and root decomposition), nitrogen cycling in soils (potential nitrification, activity of urease and denitrification enzyme, gene abundances of ammonia oxidising bacteria and archaea (*amoA* AOA and *amoA* AOB), abundances of nitrogen fixation (*nifH*) and nitrite reductase genes (*nxrA*)), functions related to the carbon cycle in soils (activity of ß-glucosidase, N-acetyl-ß-glucosaminidase, xylosidase), aboveground biomass production and total soil respiration. Most ecosystem functions data was extracted from a synthesis dataset of the Biodiversity Exploratories: ref. 241.

**Soil enzymes.** Soil samples were taken from each plot in the beginning of May 2011 and 2014 as a mixed sample from 14 soil cores of the top horizon (0–10 cm). Denitrification enzyme activity (DEA) was measured in 2011 only according to refs. 242,243. Urease activity was determined in 2011 only by incubating 1 g of fresh soil with 1.5 ml of 0.08 M substrate (urea) solution at 37 °C for 2 h[244]. Released ammonium was extracted with 12 ml of a 1 M potassium chloride/0.01 M hydrochloric solution and determined by a modified Berthelot reaction. The abundance of ammonia oxidation gene of archaea, bacteria and nitrogen fixation gene in soil bacteria different functional genes (*nifH, amoA*) is quantified via real-time qPCR analysis in both 2011 and 2014 and averaged across years. The abundance of ammonia nitrite-oxidizing Nitrobacter bacteria and archaea was estimated in 2014 using respective *amoA* gene, while NS-like and NB-like NOBs were targeted by primer sets for 16S rRNA genes for NS and *nxrA* primers genes specific for NB[245]. The abundance of nitrite-oxidizing Nitrospira was estimated using 16S rRNA gene primer specific for Nitrospira[245]. For potential nitrification, ammonium and nitrate concentrations were measured after $CaCl_2$ extraction, potential nitrification was

determined according to ref. 246 and averaged across years. Activities of the soil enzymes soil enzyme ß-glucosidase, N-acetyl-ß-glucosaminidase, and ß xylosidase were determined according to ref. 247 as described in detail in ref. 248, using fluorescent 4-methylumbelliferone substrates (4-MUF; Sigma-Aldrich, St. Louis, USA) and a buffered solution (pH 6.1).

Open datasets from Biodiversity Exploratories: refs. 249–252.

**Decomposition rates.** To measure dung decomposition rates, five dung samples were placed on each site in 2014. We used dung with a fresh weight of approx. 220.7 ($\pm$19.9) g of cow, 34.4 ($\pm$3.8) g of horse, 50.5 ($\pm$3.6) g of sheep, 32.6 ($\pm$1.6) g of deer, 14.5 ($\pm$1.4) g of fox and 47.6 ($\pm$2.4) g of wild boar. All dung samples were placed on cellulose paper. After 48 h dung samples of removal experiments were collected, transferred into small paper bags, labelled (date, site-ID, dung type) and stored in a freezer at −20 °C. Removal samples were transferred into drying ovens and kept there at 60 °C for at least five days. Afterwards the dry weight for each dung sample was weighed (Mettler Toledo "EL 2001" ($\pm$0.01 g), Columbus, Ohio) and noted for further calculations. Open dataset from Biodiversity Exploratories: ref. 253.

We measured the decomposition of fine roots (<2 mm) within the upper 10 cm of the mineral soil in 2012 with standardised herbaceous roots. Root material was left to decompose in litterbags with mesh size of 100 $\mu$m during six months (until October 2012), and mass loss was determined. Open dataset from Biodiversity Exploratories: ref. 254.

To measure litter decomposition, around 1.5 g of dry plant material collected in each plot in spring 2012 were put back in their original location in January 2013. The aboveground vegetation was removed, and bags were fixed to the ground with nails or wood sticks. There were ten replicate bags in each plot, five of them were collected two months later, and the five last ones two more months later. The dry biomass remaining in each bag was measured to calculate daily decomposition rates. Open dataset from Biodiversity Exploratories: ref. 255.

**Soil respiration.** Soil respiration was measured from late June–July in 2018 and 2019, using the soda-lime adsorption absorption method with an open and static chamber to determine soil $CO_2$ efflux. Soda-lime, i.e., mainly $Ca(OH)_2$ and NaOH, was used as the adsorption absorption material. We installed four chambers, forming a square of 10 m side length (around the weather station), and one bottom-sealed trap served as control. The mass of soda-lime for each measurement was -12 g/d and the exposure time was three days. Aboveground vegetation was carefully clipped and removed from the installation area and PVC rings was were plugged down to 1 cm soil depth. Soda-lime and lid straps were installed five to six days after vegetation clipping. Just before installation, we re-wetted the soda-lime to compensate for the initial moisture content of about 18%, since $CO_2$ needs to be hydrated before reacting with soda-lime. The $CO_2$ flux is calculated from the dry soda-lime mass differences considering i) the exposure time, ii) the area of the chamber, iii) the coefficient of 1.69 to account for the water lost during the reaction and iv) after correcting with controls (bottom-sealed) traps. The soil efflux is calculated by the equation[256,257]: Rs = [(WGsample-WGblank)/CA] × [24/T] × [12/44] × 1.69 where Rs is the soil respiration in [gCO$_2$-C m$^{-2}$ d$^{-1}$], WG is the weight gain [g], CA is the chamber basal area in [m$^2$], T is the implementation time in [h], 12/44 is the ratio of carbon atomic mass over carbon dioxide molecular mass and 1.69 compensates for the $H_2O$ formed during $CO_2$ sorption and lost during drying. Open dataset from Biodiversity Exploratories: ref. 258.

**Plant biomass production.** On two subplots of 1.5 m × 2 m size (varying position every year between 2009 and 2017), aboveground biomass was harvested between mid-May and mid-June by clipping the vegetation at a height of 5 cm. Samples were dried at 80 °C for 48 h and weighted. An arithmetic mean of biomass per m² for each plot plot was calculated. Data was then averaged across years 2009–2017. Open datasets from Biodiversity Exploratories: refs. 259–267.

### Data analysis

All the data analyses were conducted using the R software v. 4.0.3[268].

Initially, in a confirmatory analysis, species-level fast-slow axes were identified from PCA that included all species within most guilds (Fig. 2).

As we hypothesised changes in community level properties across environmental gradients (rather than changes in individual species), we focused on changes in average trait values across gradients. We thus did not consider the diversity of functional strategies in a community, or the potential for differing responses of individual species, in this analysis. Average trait values can represent both changes in species identity (taxonomic turnover) and variation in their relative abundance along the gradient. Thus, all further analyses were conducted at the level of community abundance-weighted trait mean[18] (CWM). CWM traits captures the average functional strategy of the community both in terms of response to environmental conditions (response traits) and how it reciprocally affects the functioning or other biotic components of the system (effect traits[269]). In practice, the same traits can influence both response and effect and thus can correlate strongly with both environmental drivers and ecosystem functioning[270]. CWM were calculated using relative abundance data depending on the considered guild (e.g., cover for plants, number of individuals for arthropods, number of Amplicon Sequence Variants (ASV) (fungi) or Operational Taxonomic Units (OTU) (bacteria, protists), use of the habitat and activity for birds and bats). As a result of the Biodiversity Exploratories design, data from different guilds was sometimes collected in an uncoordinated way (e.g., every year, every three years, or on fewer consecutive years). To allow comparison across guilds, when data was available for more than one year, we first calculated CWM values independently for each year, then aggregated the CWM across years. Data collected in different years, but from the same plot, were considered comparable because both land-use intensity and multivariate CWM for all guilds that were sampled more than once differed more across plots than across time. This was shown by conducting variance partitioning analyses (using the varpart function, R package vegan) with either the land-use intensity or the trait CWM value for each Plot Year combination as the response matrix, and the plot and year (as factors) as explanatory matrices. The variance explained by the plot term was on average >15 higher by that explained by the year term (Table S5). Biologically this is because high intensity fields tend to be managed at high intensity year after year, and vice-versa.

Each guild was represented by two to six traits (4.2 on average), except protists for which only one was available. Taxa with missing trait data were excluded from the corresponding CWM calculation (except in a few specific cases where data was imputed, see Methods section on functional traits); the resulting coverage (% of abundance, in terms of cover or individuals with available trait data) was always above 80% except for bacterial and Oribatid mites traits (Table S2). The CWM approach was favoured over joint species distribution modelling of trait-environment relationships[271], as it was the trait values of entire guilds and communities, not species, that was the appropriate unit of replication in our study. It was therefore essential to summarise variables at the whole guild, community, and ecosystem levels prior to analysis.

To gain a more reliable estimate of how CWM trait data was related to the hypothesised drivers, it was necessary to correct for environmental covariates before analysis. As such correction has been shown to produce biased parameter estimates in the presence of correlation between the environmental covariates[272], we identified highly correlated variables from our list of potential covariates (mean

annual temperatures and precipitation, topographic wetness index, soil clay and sand content, pH, depth and region). We excluded soil depth and sand content which were both highly anticorrelated to soil clay content ($|r| > 0.72$); and precipitation which was highly anticorrelated with temperature ($r = -0.77$). We thus retained mean annual temperature, topographic wetness index, soil clay content and pH as well as the region as covariates. To investigate the importance of these factors relative to land-use intensity, our focus variable, we fitted linear models with each trait CWM as a response and all covariates and land-use intensity as explanatory variables. This showed that on average the region explained as much variance as land-use intensity (around 8%), followed by the topographic wetness index (2%), but that there was high variability across guilds. For instance, bats and Collembola were primarily driven by the region, secondary consumer protists by soil pH, and plants or aboveground primary consumer arthropods by land-use intensity (Fig. S1). After that, we fitted linear models with each trait CWM as a response, and the covariates, excluding land-use intensity, as explanatory variables. The residuals of these linear models were used for all further analyses. Results of sensitivity analyses with uncorrected values can be found in Tables S15–S18 and Figs. S15–S20.

In cases where abundance data was not available for all 150 plots, guild-level analyses were run on all available plots (never <110 plots). To test for the response of each individual CWM trait to land-use intensity, we calculated the slope of the regression of the CWM trait against land-use intensity (excluding missing values) between each trait and land-use intensity (p-values corrected for false detection rates using the p.adjust function, $n = 47$, shown in Fig. 4). For each guild, we then sought to identify a slow-fast axis based on pre-established hypotheses based on either environmental filtering through resource availability and disturbance, or indirect, trophically-mediated mechanisms. Indeed, because traits represent consistent functional strategies, 'slow' and 'fast' traits are expected to covary. This guild-level aggregation from individual traits CWM to guild-level slow-fast axes was necessary to make the guilds comparable in the subsequent analyses. Also, this approach was chosen because the main objective of the study was not to establish the existence of such guild-level slow-fast axes, but rather examine their association at the whole community level. We ran a Principal Component Analysis (PCA) on all CWM traits of each considered guild separately. For these PCA, we used the R package factoextra, retained three principal components axes (or two if only two traits were available) and identified the PCA axis that best represented a slow-fast continuum. This was axis 1 for all guilds except the aboveground predatory arthropods (Fig. 2). Notably, the fast-slow axis was also always that which most strongly correlated with LUI, except for belowground primary consumer arthropods and birds where axes 1 and 3 (respectively) had a similarly strong correlation (Fig. S2). Guild-level PCA axes were considered to represent a slow-fast continuum (and called hereafter 'guild slow-fast axes') if they fulfilled all following conditions: (i) the considered axis explained at least twice the variance expected at random (i.e., 2/(number of traits), with 1/(number of traits) the average variance per axis expected if traits are independent); and ii) the considered axis was significantly correlated with all traits (p-value < 0.05, corrected for false detection rates, $n = 34$), with at least 60% correlations ≥ 0.4. The 'slow-fast' axis was considered only partially represented if it (i) explained at least twice the variance expected at random; (ii) was correlated ($r > 0.25$) with at least 60% of the traits, or (iii) had correlations in the unexpected direction with one trait.

For analyses across guilds, we excluded one guild (belowground primary consumer arthropods) as more than 20% of plot data were missing due to absence of this functional guild in the samples associated to some plots.

To test whether the slow-fast responses were synchronous across guilds, we ran a PCA on the previously identified slow-fast axes of all guilds, and projected the land-use intensity index (LUI) as a supplementary variable on this PCA. This approach of combining multiple PCAs into a "main" PCA follows the same logic as Multiple Factor Analysis (MFA[273,274]) and allowed us to simultaneously analyse the response of guilds with different traits and individual responses. Missing values for the slow-fast axis at the plot level (shown at the individual trait level in Table S2) were given the average of the considered guild across all plots to allow comparison across guilds (imputed values: 1.8% of all values, 0–12% range within guilds). This was done to allow comparison across guilds, and because PCA cannot be used with incomplete data. Additionally, we also tested for the existence of the whole community slow-fast axis by running a PCA on all considered traits (rather than guild-level slow-fast axes, themselves extracted from PCAs, Fig. 3b).

In follow-up analyses, we used Structural Equation Models to assess whether the ecosystem-level slow-fast continuum was driven by a common, but independent, response of individual guilds to land-use intensity, or mediated by trophic interactions between them leading to a cascade of trait matching. Because plants can affect the traits of other organisms not only through direct consumption but also by e.g., structuring the habitat[275], we also included paths between all guilds and plants. We also allowed for some correlation paths between guilds when it made biological sense and significantly improved model fit (e.g., between birds and bats, which might be jointly affected by landscape-scale variables). Models were fitted using the lavaan R package[276], separately for below and aboveground guilds; plants were included in both. The hypothesised model structures are shown in Fig. S3 and model statistics in Tables S6 and S7.

The SEM were fitted using the maximum likelihood estimator using all available data (i.e., excluding NAs/imputed data points but using all existing data for each estimate). Estimates were bootstrapped 200 times. To assess how path strength varied across trophic levels, we extracted the coefficients and corresponding confidence intervals for all direct, indirect and total effects of land-use intensity on each trophic level. We then fitted linear models with estimated direct, indirect and total effects of LUI as the response variables and trophic level as the explanatory variable. Effect estimates were weighted by the inverse of the standard error to account for variable uncertainties across guilds. To evaluate the overall ecosystem response to LUI, we averaged the direct, indirect and total effects at each trophic level, independently above- and belowground. This was done by defining custom parameters within the SEM as the average of all corresponding parameters (direct/indirect/total for all guilds within each trophic level). This allowed the same bootstrap procedure to also estimate average effects and their confidence intervals.

Finally, we investigated the possible linkage between the entire community slow-fast axis and a potential ecosystem functioning slow-fast axis. We selected 15 ecosystem function indicators from five bundles of related functions (see above). All functions were corrected for environmental variables (as described above for traits) before analysis. All selected functions were hypothesised to represent 'fast' ecosystem functioning (e.g., fast nutrient cycling) at high values.

We first investigated the correlation between individual traits and functions; or guild slow-fast axes and function bundles (Figs. S5 and S6). We then conducted a PCA on all functions, weighted so that each functional bundle would have the same final weight (e.g., functions related to N fluxes were weighted 1/8, decomposition functions were weighted 1/3, etc.). This was done to prevent bundles with more functions (e.g. N fluxes) from having a disproportionate impact upon the PCA. The first axis (29% of total variance) represented a continuum from slow to fast nutrient cycling with high potential nitrification, and high activities of most enzymes. The plot-level PC scores of this ecosystem functions axis were then regressed against the 'slow-fast' axes of individual guilds (microbes and plants,

expected to be the main driver of soil functioning), the overall ecosystem functional trait slow-fast axis, land-use intensity and multidiversity, a measure of the overall species diversity of all considered groups (Table S4). This was calculated as the average richness of all taxonomic groups, scaled by the total species richness of each group[32], and also corrected for environmental variables, as described above. Finally, to test the direct and indirect effect of land-use intensity on the slow-fast ecosystem functioning axis, we fitted a SEM (using the lavaan package) testing the indirect effect via the entire community slow-fast axis (Fig. 6b).

Additionally, we tested for the direct and indirect effects of land-use intensity on the ecosystem slow-fast axis, but on the community slow-fast axis that was derived from the 'all traits' PCA (rather than guild-level slow-fast axes extracted from individual PCAs, Fig. S4).

### Reporting summary
Further information on research design is available in the Nature Portfolio Reporting Summary linked to this article.

## Data availability
This work is based on data collected within several projects of the Biodiversity Exploratories program (DFG Priority Program 1374). Most datasets from the Biodiversity Exploratories are publicly available in the Biodiversity Exploratories Information System (Bexis) (https://doi.org/10.17616/R32P9Q). The CWM data generated in this study have been deposited under accession code 31689 https://www.bexis.uni-jena.de/ddm/data/Showdata/31689. The raw trait and abundance, and ecosystem functions datasets are listed below, most of which are publicly available. To give data owners and collectors time to perform their analysis the Biodiversity Exploratories' data and publication policy includes by default an embargo period of three years from the end of data collection/data assembly. Access to the remaining datasets can thus be obtained by contacting the Biodiversity Exploratories office or data owners. At the end of the embargo period these datasets will be made publicly available via the same data repository. Full list of used datasets (both from the Biodiversity Exploratories and external, previously published datasets)[3,19,45,47,53–56,93–97,101–119,221–235,240,249–254,268–109,110,277].

## Code availability
Full code to replicate the analyses is stored on GitHub at https://github.com/mneyret/trait_synchronies under the https://doi.org/10.5281/zenodo.10286643.

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

## Acknowledgements

The work was partly funded by the DFG Priority Program 1374 'Infrastructure-Biodiversity- Exploratories', and by Senckenberg Biodiversity and Climate Research Center. The soil analyses (EK, SM) was funded by the DFG (KA 1590/8-5, MA 4436/1-5). DB, ASZ, KB, KJ and VW have received funding in the context of Resilience DFG project 413730012. ASZ was supported by the RSF 23-14-00201 grant. Fieldwork permits were issued by the state environmental offices of Baden-Württemberg, Thüringen, and Brandenburg. The authors thank Konstans Wells, Kirsten Reichel-Jung, Sonja Gockel, Kerstin Wiesner, Katrin Lorenzen, Andreas Hemp, Martin Gorke maintained the plot and project infrastructure; Simone Pfeiffer, Maren Gleisberg, Christiane Fischer, Jule Mangels and Victoria Grießmeier for administrative support, and Jens Nieschulze, Michael Owonibi and Andreas Ostrowski for database management. Eduard Linsenmair, Dominik Hessenmöller, Daniel Prati, Ernst-Detlef Schulze, Wolfgang W. Weisser and Elisabeth Kalko helped establish the Biodiversity Exploratories project. The administration of the Hainich National Park, the UNESCO Biosphere Reserves Swabian Alb and Schorfheide-Chorin and all land owners provided logistical support. Tom Lachaise, Jörg Overmann and Johannes Sikorski contributed data. Icons were acquired and adapted (colour, width-height ratio) from Phylopic.org. Lepidoptera, fungi, secondary consumer arthropods, bats, protists, Oribatid mites: public domain. Birds: Maxime Dahirel (CC BY 3.0 Deed). Collembola: Birgit Lang (CC BY 3.0 Deed). Bacteria: Matt Crook (CC BY-SA 3.0 Deed). Hare: Jan A. Venter, Herbert H. T. Prins, David A. Balfour & Rob Slotow (vectorised by T. Michael Keesey) (CC BY 3.0 Deed). Tortoise: Andrew A. Farke, shell lines added by Yan Wong (CC BY 3.0 Deed). Primary consumer arthropods: Gareth Monger (CC

BY 3.0 Deed). Link to the corresponding licences: CC BY 3.0 Deed: https://creativecommons.org/licenses/by/3.0/. CC BY-SA 3.0 Deed: https://creativecommons.org/licenses/by-sa/3.0/.

## Author contributions

M.N., F.D.S. and P.M. designed the study. M.N. and P.M. conducted the analyses with inputs from G.L.P. and A.L.B. M.N., P.M., G.L.P., D.B., F.T.d.V., J.B., A.M.F.D., S.G., K.G., A.M., R.A.S., N.K.S., J.A.T., A.S.Z. participated to workshops and expert discussions to define guild-specific trait lists and hypotheses. M.N. and P.M. wrote the manuscript with significant inputs from G.L.P., A.L.B., D.B., J.B., S.G., J.A.T., M.M.G., E.K., K.B., K. Jung, J. K., C.P., M.C.R., M. Schlöter, S. Schulz, M. Staab, M.v.K. and V.W. G.L.P., D.B., J.B., A.M.F.D., K.G., R.S., N.S., J.T., A.Z., M.G., K. John, K. Jung, E.K., J.K., C.P., M. Schloter, S. Schulz, M. Staab, V.W., A.A., S.B., R.S.B., R.B., M.B., F.B., K.D., H.Y.G., N.H., J.H., K.J., V.H.K, T.K., S.M., J.M., S.C.R., N.V.S., I.S., M.S., S. Seibold, S.A.S., E.S., M.T., M.v.K., T.W., M.F. and P.M. contributed data. All authors commented on the manuscript. Author order was determined as follow: main authors, workshop and discussion participants (alphabetical), other authors with significant input (alphabetical), other data contributors (alphabetical), senior author.

## Funding

## Competing interests

The authors declare no competing interests.

## Additional information

Margot Neyret [1,2] ✉, Gaëtane Le Provost [3], Andrea Larissa Boesing[1], Florian D. Schneider [1,4], Dennis Baulechner [5], Joana Bergmann [6], Franciska T. de Vries [7], Anna Maria Fiore-Donno [8], Stefan Geisen[9], Kezia Goldmann [10], Anna Merges[1], Ruslan A. Saifutdinov[11], Nadja K. Simons[12,13], Joseph A. Tobias [14], Andrey S. Zaitsev [5,11,15], Martin M. Gossner [16,17], Kirsten Jung [18], Ellen Kandeler[19], Jochen Krauss [20], Caterina Penone [21], Michael Schloter[22,23], Stefanie Schulz [22], Michael Staab [12], Volkmar Wolters [5], Antonios Apostolakis [24,25], Klaus Birkhofer[26], Steffen Boch [27], Runa S. Boeddinghaus [19,28], Ralph Bolliger [21], Michael Bonkowski [8], François Buscot [10,29], Kenneth Dumack [8], Markus Fischer [21], Huei Ying Gan[30], Johannes Heinze [31], Norbert Hölzel [32], Katharina John[5], Valentin H. Klaus [33,34], Till Kleinebecker [35,36], Sven Marhan[19], Jörg Müller[37], Swen C. Renner[38], Matthias C. Rillig [39], Noëlle V. Schenk [21], Ingo Schöning [24], Marion Schrumpf [24], Sebastian Seibold [40,41], Stephanie A. Socher [42], Emily F. Solly [43], Miriam Teuscher[44], Mark van Kleunen [45,46], Tesfaye Wubet [29,47] & Peter Manning [1,48] ✉

[1]Senckenberg Biodiversity and Climate Research Centre, Frankfurt, Germany. [2]Laboratoire d'Écologie Alpine, Université Grenoble Alpes - CNRS - Université Savoie Mont Blanc, Grenoble, France. [3]INRAE, Bordeaux Sciences Agro, ISVV, SAVE, Villenave d'Ornon, France. [4]ISOE - Institute for social-ecological research, Frankfurt am Main, Germany. [5]Justus Liebig University, Department of Animal Ecology, Giessen, Germany. [6]Leibniz Center for Agricultural Landscape Research (ZALF), Müncheberg, Germany. [7]Institute for Biodiversity and Ecosystem Dynamics, University of Amsterdam, Amsterdam, The Netherlands. [8]Terrestrial Ecology, Institute of Zoology, University of Cologne, Köln, Germany. [9]Laboratory of Nematology, Wageningen University and Research, Wageningen, The Netherlands. [10]Helmholtz Centre for Environmental Research (UFZ), Soil Ecology Department, Halle/Saale, Germany. [11]A.N. Severtsov Institute of Ecology and Evolution, Russian Academy of Sciences, Moscow, Russia. [12]Ecological Networks, Technical University Darmstadt, Darmstadt, Germany. [13]Applied Biodiversity Sciences, University of Würzburg, Würzburg, Germany. [14]Department of Life Sciences, Imperial College London, Ascot, UK. [15]Senckenberg Museum for Natural History Görlitz, Görlitz, Germany. [16]Forest Entomology, Swiss Federal Research Institute WSL, Birmensdorf, Switzerland. [17]Department of Environmental Systems Science, Institute of Terrestrial Ecosystems, ETH Zürich, Zürich, Switzerland. [18]Institut of Evolutionary Ecology and Conservation Genomics, Ulm University, Ulm, Germany. [19]Department of Soil Biology, Institute of Soil Science and Land Evaluation, University of Hohenheim, Stuttgart, Germany. [20]Department of Animal Ecology and Tropical Biology, Biocenter, University of Würzburg, Würzburg, Germany. [21]Institute of Plant Sciences, University of Bern, Bern, Switzerland. [22]Helmholtz Zentrum Muenchen, Research Unit for Comparative Microbiome Analysis, Oberschleissheim, Germany. [23]Chair of Environmental Microbiology, Technical University of Munich, Freising, Germany. [24]Department of Biogeochemical Processes, Max-Planck-Institute for Biogeochemistry, Jena, Germany. [25]Department of Crop Sciences, University of Göttingen, Göttingen, Germany. [26]Department of Ecology, Brandenburg University of Technology Cottbus-Senftenberg, Cottbus, Germany. [27]Swiss Federal Research Institute WSL, Birmensdorf, Switzerland. [28]Department Plant Production and Production Related Environmental Protection, Center for Agricultural Technology Augustenberg (LTZ), Karlsruhe, Germany. [29]German Centre for Integrative Biodiversity Research (iDiv) Halle - Jena-, Leipzig, Germany. [30]Senckenberg Centre for Human Evolution and Palaeoenvironments Tübingen (SHEP), Tübingen, Germany. [31]Department of Biodiversity, Heinz Sielmann Foundation, Wustermark, Germany. [32]Institute of Landscape Ecology, University of Münster, Münster, Germany. [33]Institute of Agricultural Sciences, ETH Zürich, Zürich, Switzerland. [34]Forage Production and Grassland Systems, Agroscope, Zürich, Switzerland. [35]Institute for Landscape Ecology and Resources Management (ILR), Research Centre for BioSystems, Land Use and Nutrition (iFZ), Justus Liebig University Giessen, Giessen, Germany. [36]Centre for International Development and Environmental Research (ZEU), Justus Liebig University Giessen, Giessen, Germany. [37]Department of Nature Conservation, Heinz Sielmann Foundation, Wustermark, Germany. [38]Ornithology, Natural History Museum Vienna, Vienna, Autria, Germany. [39]Freie Universität Berlin, Institute of Biology, Berlin, Germany. [40]Technical University of Munich, TUM School of Life Sciences, Freising, Germany. [41]TUD Dresden University of Technology, Forest Zoology, Tharandt, Germany. [42]Paris Lodron University Salzburg, Department Environment and Biodiversity, Salzburg, Austria. [43]Helmholtz Centre for Environmental Research (UFZ), Computation Hydrosystems Department, Leipzig, Germany. [44]University of Göttingen, Centre of Biodiversity and Sustainable Land Use, Göttingen, Germany. [45]Zhejiang Provincial Key Laboratory of Plant Evolutionary Ecology and Conservation, Taizhou University, Taizhou, China. [46]Ecology, Department of Biology, University of Konstanz, Konstanz, Germany. [47]Helmholtz Centre for Environmental Research (UFZ), Community Ecology Department, Halle/Saale, Germany. [48]Department of Biological Sciences, University of Bergen, Bergen, Norway. ✉e-mail: margot.neyret@univ-grenoble-alpes.fr; peter.manning@uib.no

