## [Peer Review File · Nature Communications]

Land use intensification drives a slow-fast trait continuum at the level of entire communitiesEditorial Note: This manuscript has been previously reviewed at another journal that is not operating a transparent peer review scheme. This document only contains reviewer comments and rebuttal letters for versions considered at Nature Communications.

Reviewers' Comments:

Reviewer #3:

Remarks to the Author:

I have previously reviewed this manuscript (as reviewer 3), which tests hypothesis regarding how communities shift along a slow-fast continuum in response to land use intensification. As I mentioned in the previous review this is an interesting and important question and an ambitious attempt to address it using extensive data and a series interwoven models. I must strongly commend the authors for address many of the previous comments with an extensive re-writing of the manuscript and the inclusion of additional analysis. In particular, they have addressed my main concern regarding the role of body mass in their overall analysis. While the results may be weaker when mass is removed this is certainly something to be expected but shows that the main results are not just the result of some form of allometric scaling but are also associated with classic slow-fast continuum trade-offs. I also think the authors have made the hypothesis and statistical approaches clearer in this manuscript. I do agree with other reviewers that the stacked modelling approach used can feel a little uneasy in terms of how variation is lost across each step and the relatively low tractability of the approach. However, I do also think that such adventurous models are required in order to work at the scale of the question and are a useful addition to the field.

Reviewer #4:

Remarks to the Author:

Comments to the authors:

I was not the original reviewer of this manuscript, but have been asked to comment specifically on several points raised in the previous review. One of those points was regarding the use of CWMs, which I will address here first. The original reviewer suggesting that "Careful revision of the intentions of the study are needed to make a stronger case for the use of CWM as a metric here." My main interpretation of this comment was that 'it does not capture any trade-offs between functions'. This is true.

The author's reply does not specifically address this later point (i.e. trade-offs) in the justification for using CWM trait values. Rather the authors cite an article by Muscarella and Uriarte (2016) suggesting that CWMs reflect the optimal trait value of a site. To be honest, I was unfamiliar with the cited article and only briefly examined it, but I would strongly caution the authors on this justification/rationale within their introduction. I also not think this is necessary. The authors state (citing M&U 2016): "species with traits closer to the CWM value at any given site should be those with highest fitness and thus highest abundances". This is circular. CWMs are calculated from the proportional species abundances at the locality. So that the species with the higher weighting (i.e. higher prop abundance) drive the CWM value found at the local site. How could there be any other relationship? – a CWM by definition is the trait mean weighted by proportional abundance. Perhaps this is not what the authors of the original cited article (M&U 2016) show, but regardless, I am not convinced that CWMs confer any type of 'optimality' of the organisms. I would agree that strong selective pressures can shape or alter the CWM value of a trait (e.g. drought tolerance trait under a drought treatment), but this is a response to a perturbation and not an 'optimal' life history strategy for an environment per se.

Back to the main point however, a justification to examining CWMs, I do not see the need to invoke a

hypothesis of 'optimality' associated with changing CWM trait values, and it does not address ideas of trade-offs or alternative life history strategies. I suggest removing paragraph lines 114-135 in the introduction, and move this to a brief rationale for CWM in the methods section, and the rest in the discussion to address reviewer's original comment. More so, I think what is important to consider and state explicitly is that the CWM is a derived and calculated variable, AND that the trait itself may be a response or an effect trait. Differentiating this is what's important.

The second part of the reply to the original reviewer comment on CWMs is in the discussion. I also agree with the points raised here by the authors, but again I think perhaps the original reviewer point is missed, i.e. an analysis of CWMs does not capture potential trade-offs in life history strategies or species response to perturbation, but the discussion provided by the authors comes closer.

I also have a few independent comments on this manuscript.

First, I commend the authors for approaching this topic. The fast-slow spectrum is well recognized, but rarely synthesised across both aboveground and belowground components of ecosystems. That said, I was disappointed by the lack of specific details on belowground aspects of this. Further to this, Table S14 suggests that the response of almost all belowground organisms with respect to fast-slow traits were inconclusive.

I was also surprised that despite the results that body size was smaller for many groups under perturbation (Fig 3) the authors fail to invoke the concept of 'community downsizing' that has been well established under perturbation, and specifically climate change.

However, I am always sceptical when the supplementary information is longer than the manuscript and includes extensive methods and results that were used to perform subsequent analyses. Figure 2 is an example of that manifestation. It is unclear in the main text how Figure 2 is created. The Caption does not provide much information nor does it reflect what the intention of the study was or how this figure supports the overall outcome of the study. The caption states "The variables included in the PCA are the slow-fast axes of each guild", but what does this mean? As I understand this from the methods (at the end of the manuscript – not my favourite format for just this reason), this is the PCA axis scores for each group that the authors deemed associated with fast-slow traits (line 695). To understand this, one has to go to the 95 pages (not including references) of the supplementary material. This is where I am not sure where to look for the results that support these results (Table S2 perhaps, or Table S8, or Table S13?). But to my understanding, Fig 2 is showing the response in fast-slow traits across all groups, but this is artefactual. These vectors were already selected because they show what the authors are looking for (i.e. a fast-slow spectrum), thus Figure 2 is not surprising as it is a product of the authors finding the best vector to reflect fast-slow. Again, seems a circular argument here; selecting the fast-slow axis of each group to show there is a fast-slow axis.

Ultimately, while I support the endeavour of this manuscript, I felt real mechanism was missing linking fast-slow traits to fast-slow ecosystem system processes, and the discussion was too general in nature to be useful.

Reviewer #5:

Remarks to the Author:

This paper investigates the effect of land use intensity on community-level mean values of traits that reflect a continuum of functional strategies from slow- to fast growing organisms. The authors analyzed an impressive occurrence and trait database and identified the main slow-fast functional trait axes in 14 trophic guilds. They show direct and indirect effects of land use intensity on the slow-fast

trait axes. They also showed that the slow-fast trait axes were associated with the rates of whole ecosystem functioning. The authors conclude by discussing the potential of the observed 'slow-fast' continuum for whole ecosystem-level studies.

I very much enjoyed reading this manuscript which has a clear structure and is well written. The topic of the study fits within the scope of the journal. The main questions are relevant and timely. Overall, the work and methods meet the expected scientific standards. The conclusions are supported by the results. The figures and tables are clear and of good quality. The literature cited is appropriate.

The study summarizes complex processes in a simple message that can help improve future ecosystem conservation and management efforts. I think the study will be of interest to a broad range of researchers and practitioners in the field of ecology and beyond. I have not found any major weaknesses or errors in this study. However, I make several suggestions for possible minor improvements.

Note that, as requested, my review mostly focuses on the data analysis part of the manuscript. See my comments below for further details.

Main comments:

1. Trait selection

Although the authors did considerable efforts to explain this process, I think that the interpretation of PCA axes as slow-fast and their selection for further analyses can be seen as a weak point. For example, I agree with R3 that important traits in bats and birds are missing and would be easy to add given the high amount and quality of trait information for these two taxa. Furthermore, if traits were originally selected to represent the slow-fast continuum, why select just one PCA axis? The non-selected axes might represent other important ecological strategies. Traits related to reproduction might have different drivers than those related to growth. Some aspects of the slow-fast trait continuum might not be driven by land use intensity. To me, this selection process, especially, the selection of PCA axes, casts some doubts on the validity and generality of the conclusions. However, the authors did provide a complementary analysis based on all traits (Fig.S3) where the slow-fast axis emerges in a more direct and organic way. This additional analysis alongside Fig.3 did put my initial concerns at ease as the results show that the slow-fast axis is present even without selecting PCA axes. I encourage the authors to put more emphasis on these two analyses in the text. For example, the authors could mention Fig.S3 directly in the caption of Fig.2. See also my comments about fourth corner/RLQ below.

2. Controlling for the effect of environmental covariates

Controlling for the effect of environmental covariates is indeed a necessary step and I agree with the approach used by the authors to address this issue. However, I miss summary information about this step. How much variation is explained in each taxon? Do we have large differences among taxa? Which variables explain the most variation? After my first read, I was wondering whether this step can cause some biases. However, Table S14 shows correlations between LUI and CWM where CWMs were not corrected for environmental covariates and the results are consistent with the other analyses. This suggests that the results are robust.

I think that RLQ would be interesting to characterize the links between traits and environmental variables. This analysis would provide important information about the strength of the effect of LUI on traits as compared to that of other covariates. It would also provide a more general picture of the ecology of the different taxonomic groups. Whether or not the authors decide to use the RLQ, I think

that more information is needed about the relative importance of the influence of LUI on traits as compared to that of other environmental variables.

Dray, S., Choler, P., Dolédec, S., Peres-Neto, P. R., Thuiller, W., Pavoine, S., & ter Braak, C. J. (2014). Combining the fourth-corner and the RLQ methods for assessing trait responses to environmental variation. *Ecology*, 95(1), 14-21. DOI 10.1890/13-0196.1

3. Stacking of models

I agree with the main logic behind the multiple PCAs. Basically, it is the same idea as in a multiple factor analysis, MFA (see refs below). It is, to me, a valid and interesting way to conduct a multi-taxa analysis. However, I also agree with R2 that the analysis procedure is relatively complex which makes it difficult to keep track of the potential biases and information loss at each step. All in all, I see no major problem with the stacking of models and encourage the authors to cite the MFA papers as a reference framework for their approach.

Escofier, B. and Pages, J. (1994) Multiple Factor Analysis (AFMULT package). *Computational Statistics and Data Analysis*, 18, 121-140.

Becue-Bertaut, M. and Pages, J. (2008) Multiple factor analysis and clustering of a mixture of quantitative, categorical and frequency data. *Computational Statistic and Data Analysis*, 52, 3255-3268.

4. Trait-function relationship

Although not essential to the story, I miss the direct relationship between traits and EF. Here, the fourth-corner/RLQ analysis could also be used to test the associations between individual traits and EF variables. I think that such analyses would constitute a complementary and more direct approach to highlight the slow-fast trait continuum. I expect that this analysis will reveal significant correlations that will further validate the other analyses and increase the confidence in the results.

Minor comments:

The use of CWM: Whether CWM reflects optimal strategies has been subject to much discussion. I personally think that this is not always the case. For instance, a bimodal distribution of trait values within a community can indicate the coexistence of two successful strategies. In such a case, the mean value would reflect a sub-optimal strategy. Having said that, I do not think that such bimodal trait distributions are highly frequent and this should not be an issue for the validity of the results. I, nevertheless, encourage the authors to add a few words about this.

It was difficult for me to understand how was Fig.3 done. I encourage the authors to provide a few additional explanations in the text and directly in the caption.

Figure 4: Having two slightly different color scales for model estimates is confusing. Would it be possible to use one single scale for both panels?

Table S6: What is the shared variation?

Overall, this is a very nice work and I hope that my comments will help to improve it further.

dear Reviewers,

Thank you for the opportunity to revise this manuscript and for your careful reviews. In response to your feedback we have made numerous changes to the manuscript. The most important of these are:

- To provide more transparent and interpretable information on the guild-level slow-fast axes, we now provide a new extended data figure (and reference in the text) of the guild-level PCA from which the slow-fast axes originate. These were previously in a table in the supplementary material and are now more prominent in the main text. We now also highlight the PCA conducted on all traits (rather than guild slow-fast axes) by promoting it to extended data figure 2. This helps support our interpretation of a whole community-level slow-fast axis.
- We moved the methods tables from the supplementary material to the extended data tables to improve clarity (if this is possible for the journal format).
- We ran additional analyses regarding the role of environmental covariates across the different guilds, and the relationship between individual traits and functions; we reference them in the text and provide them as supplementary material.
- We added additional discussion and clarification to the text of the main paper and methods relating to several points raised by the reviewers

We have done our best to accommodate suggestions from this round of review while still retaining revisions made in response to earlier reviews, as the requests of reviewers sometimes diverged slightly (e.g. on whether to include the introductory paragraph on CWM; and whether to include extensive sensitivity analyses in the supplementary material). We also opted to keep the addition of new material to a minimum given the complexity of the paper and already large supplementary materials (as pointed out by reviewer 4). We therefore highlighted existing materials, or expanded these slightly to address reviewer concerns rather than adding additional complex analyses.

The lines indicated in this response letter correspond to the line numbers in the document with change visible.

We feel that these changes have significantly improved the manuscript and hope you will find them satisfactory.

Margot Neyret, on behalf of all authors

REVIEWER COMMENTS

Reviewer #3 (Remarks to the Author):

3.1. I have previously reviewed this manuscript (as reviewer 3), which tests hypothesis regarding how communities shift along a slow-fast continuum in response to land use intensification. As I mentioned in the previous review this is an interesting and important question and an ambitious attempt to address it using extensive data and a series of interwoven models. I must strongly commend the authors for address many of the previous comments with an extensive re-writing of the manuscript and the inclusion of additional analysis. In particular, they have addressed my main concern regarding the role of body mass in their overall analysis. While the results may be weaker when mass is removed this is certainly something to be expected but shows that the main results are not just the result of some form of allometric scaling but are also associated with classic slow-fast continuum trade-

offs. I also think the authors have made the hypothesis and statistical approaches clearer in this manuscript. I do agree with other reviewers that the stacked modelling approach used can feel a little uneasy in terms of how variation is lost across each step and the relatively low tractability of the approach. However, I do also think that such adventurous models are required in order to work at the scale of the question and are a useful addition to the field.

R3.1 Thanks again for your previous constructive suggestions, and for your positive feedback on our revision.

Reviewer #4 (Remarks to the Author):

Comments to the authors:

4.1 I was not the original reviewer of this manuscript, but have been asked to comment specifically on several points raised in the previous review. One of those points was regarding the use of CWMs, which I will address here first. The original reviewer suggesting that “Careful revision of the intentions of the study are needed to make a stronger case for the use of CWM as a metric here.” My main interpretation of this comment was that ‘it does not capture any trade-offs between functions’. This is true.

The author’s reply does not specifically address this later point (i.e. trade-offs) in the justification for using CWM trait values. Rather the authors cite an article by Muscarella and Uriarte (2016) suggesting that CWMs reflect the optimal trait value of a site. To be honest, I was unfamiliar with the cited article and only briefly examined it, but I would strongly caution the authors on this justification/rationale within their introduction. I also not think this is necessary. The authors state (citing M&U 2016): “species with traits closer to the CWM value at any given site should be those with highest fitness and thus highest abundances”. This is circular. CWMs are calculated from the proportional species abundances at the locality. So that the species with the higher weighting (i.e. higher prop abundance) drive the CWM value found at the local site. How could there be any other relationship? – a CWM by definition is the trait mean weighted by proportional abundance. Perhaps this is not what the authors of the original cited article (M&U 2016) show, but regardless, I am not convinced that CWMs confer any type of ‘optimality’ of the organisms. I would agree that strong selective pressures can shape or alter the CWM value of a trait (e.g. drought tolerance trait under a drought treatment), but this is a response to a perturbation and not an ‘optimal’ life history strategy for an environment per se.

Back to the main point however, a justification to examining CWMs, I do not see the need to invoke a hypothesis of ‘optimality’ associated with changing CWM trait values, and it does not address ideas of trade-offs or alternative life history strategies. I suggest removing paragraph lines 114-135 in the introduction, and move this to a brief rationale for CWM in the methods section, and the rest in the discussion to address reviewer’s original comment. More so, I think what is important to consider and state explicitly is that the CWM is a derived and calculated variable, AND that the trait itself may be a response or an effect trait. Differentiating this is what’s important.

The second part of the reply to the original reviewer comment on CWMs is in the discussion. I also agree with the points raised here by the authors, but again I think perhaps the original reviewer point is missed, i.e. an analysis of CWMs does not capture potential trade-offs in life history strategies or species response to perturbation, but the discussion provided by the authors comes closer.

R4.1. Thank you for your feedback. We retained a paragraph on CWM in the introduction to remain consistent with previous reviewers’ requests, and because it is a key concept in the

paper, but have revised it to remove the initial point on optimality. The full paragraph in the intro now reads:

I. 115. “At the level of guilds and communities, winning strategies can be seen as manifesting as an emergent property, and represented in community-level trait measures, typically the community abundance weighted trait mean (Bruehlheide et al., 2018; Garnier et al., 2004) (CWM). While there can be significant trait variation within a community, a CWM captures the average functional strategy of the community; and changes in CWM across space and time reflect both turnover in species with different trait values, and variation in their relative abundance, in response to changes to the species pool and environmental conditions. The combined response of multiple trait CWMs thus represents a change in the overall functional strategy at a community level. This means that slow-fast strategy responses – encompassing a range of traits – may emerge at the community level from a concurrent change in individual CWM traits related to ‘slow’ and ‘fast’ strategies. In particular, communities of resource-acquisitive, fast-growing organisms with numerous offspring, a fast pace of life and good dispersal abilities tend to be found in resource-rich and disturbed habitats, while resource-conservative, slow-growing organisms with longer life span and fewer offspring tend to be favoured in undisturbed or resource-poor habitats (Börschig et al., 2013; Daou et al., 2021; de Vries et al., 2006; Pedley and Dolman, 2014; Simons et al., 2016). »

We also moved/expanded some of the justification and limits in the methods in relation to response/effect traits.

I. 728: “As we hypothesised changes in community level properties across environmental gradients (rather than changes in individual species), we focused on changes in average trait values. We thus did not consider the diversity of functional strategies in a community, or the potential for differing responses of individual species, in this analysis. Average trait values can represent both changes in species identity (taxonomic turnover) and variation in their relative abundance along the gradient. Thus, all further analyses were conducted at the level of community abundance-weighted trait mean (Garnier et al., 2004) (CWM). CWM traits capture the average functional strategy of the community, both in terms of response to environmental conditions (response traits) and how it affects the functioning or other biotic components of the system (effect traits, Lavorel and Garnier, 2002). In practice, the same traits can influence both response and effect and thus can correlate strongly with both environmental drivers and ecosystem functioning (Suding et al., 2008). CWM were calculated using relative abundance data [...]”

We did not understand the original point on trade-offs as we are looking at multiple traits and so we would detect trade-offs between these if they exist at the community level, our focal level of study.

4.2. I also have a few independent comments on this manuscript.

First, I commend the authors for approaching this topic. The fast-slow spectrum is well recognized, but rarely synthesized across both aboveground and belowground components of ecosystems. That said, I was disappointed by the lack of specific details on belowground aspects of this. Further to this, Table S14 suggests that the response of almost all belowground organisms with respect to fast-slow traits were inconclusive.

R4.2. Thank you. Indeed, in our system the below-ground groups appear to show a weaker response than above-ground groups. We think this can be explained by two main factors: i, our main driver (land use intensity) affects resource availability in the soil through fertilization; but the disturbance aspects of this intensification, which we expect to be an important driver of many traits, including body size, mostly occur above-ground through mowing and grazing, with a more limited impact on below-ground organisms; ii, less available trait data and taxonomic information for below- than above-ground groups led to higher-level data aggregation and this might have weakened our capacity to detect a response. We now discuss it more directly in the discussion:

I. 524: “Below-ground organisms also tended to show weaker responses than above-ground ones, likely due lower trait data availability weakening our capacity to detect a response and lower disturbance of the soil environment than the aboveground environment by the intensification studied here- mowing and grazing (Le Provost et al., 2021).”

4.3. I was also surprised that despite the results that body size was smaller for many groups under perturbation (Fig 3) the authors fail to invoke the concept of ‘community downsizing’ that has been well established under perturbation, and specifically climate change.

R4.3. Thank you for mentioning community downsizing. From what we found in the literature, this concept seems to be mostly used to describe changes in response to climate change or defaunation of large animals, without a clear link to the resource availability and disturbance gradients we focus on, so we did not want to make it too central a point in our manuscript. However, we feel this could be an important avenue for future integration of our study to broader ecological theories (e.g. to link between slow-fast continuum and size in different climate change contexts) and we now mention it in the discussion.

I. 592: “Such synchrony may also correspond to recently described global trends in the covariance of multiple ecosystem functions (Gounand et al., 2020; Migliavacca et al., 2021). It could also be related to other widely reported patterns of ecological change, such as the global “community downsizing” of animal communities that occurs in response to both climate change and anthropogenic disturbance (Lindo, 2015, Martins et al. 2023) and its impact of ecosystem functioning (Donoso et al., 2020), though further theoretical integration would also be required.”

4.4. However, I am always skeptical when the supplementary information is longer than the manuscript and includes extensive methods and results that were used to perform subsequent analyses. Figure 2 is an example of that manifestation. It is unclear in the main text how Figure 2 is created. The Caption does not provide much information nor does it reflect what the intention of the study was or how this figure supports the overall outcome of the study. The caption states “The variables included in the PCA are the slow-fast axes of each guild”, but what does this mean? As I understand this from the methods (at the end of the manuscript – not my favourite format for just this reason), this is the PCA axis scores for each group that the authors deemed associated with fast-slow traits (line 695). To understand this, one has to go to the 95 pages (not including references) of the supplementary material. This is where I am not sure where to look for the results that support these results (Table S2 perhaps, or Table S8, or Table S13?). But to my understanding, Fig 2 is showing the response in fast-slow traits across all groups, but this is artefactual. These vectors were already selected because they show what the authors are looking for (i.e. a fast-slow spectrum), thus Figure 2 is not surprising as it is a product of the authors finding the best vector to reflect fast-slow. Again, seems a circular argument here; selecting the fast-slow axis of each group to show there is a fast-slow axis.

R4.4. We fully agree that our supplementary material is a bit long, partly due to the addition of sensitivity analyses requested by previous reviewers. To emphasize the most important information within these appendixes, we would like to present some methods tables as extended data objects. If this is only possible for figures, we would place Table 1 in the main paper and keep Tables 2-5 in the SI as we feel their size will obscure the main sections and messages of the paper.

We also moved key intermediate analyses (e.g. group-level PCAs) to the main text as a new Extended Data Figure 1. The caption of Figure 2 was also updated to refer more clearly to the original PCAs shown in Extended Data Figure 1:

Figure 2. The slow-fast trait axis of individual guilds is strongly related to land-use intensity. The variables included in the PCA are the slow-fast axes of each guild, which were

estimated as the PC axis that best represents the hypothesized slow-fast axis (PC1 in 90% of cases) in a PCA of the selected CWM traits expected to be related to slow-fast strategies for each guild individually. These guild-level PCA can be found in Extended data Figure 1. Land-use intensity (which is projected on the PC axes, shown in red) is strongly associated with Axis 1. Belowground guilds are shown in brown, aboveground guilds in blue, and plants in green. Sample size: 150 (sample sizes for each individual group are shown in Fig. 3, missing values were imputed to run the PCA, see methods). An equivalent analysis, but conducted on all traits without guild-level aggregation is shown as Extended data Figure 2.

Regarding the choice of slow-fast traits leading “circularly” to group-level fast-slow axes, we would like to note that our objective was not really to demonstrate the existence of fast-slow axes for individual groups (in which case we agree the argument would be circular) but to assess whether these fast-slow axes were strongly associated at the community level – in which case we believe that our analysis is appropriate. The aggregation steps from individual traits CWM to group-level slow-fast axes was necessary to make the groups comparable in the SEM. This is now better highlighted in the introduction and methods. To support this argument, and to make our rationale clearer we now moved a supplementary figure to the main paper as Extended data figure 2 and reference it prominently in the main text (caption of Figure 2, sensitivity analyses section). This shows a PCA conducted not on group slow-fast axes but on individual traits (ie without the aggregation step to guild slow-fast axes), thus supporting the existence of a whole-community slow-fast axis.

I. 807: “For each guild, we then sought to identify a slow-fast axis based on pre-established hypotheses based on either environmental filtering through resource availability and disturbance, or indirect, trophically-mediated mechanisms (Extended data Figure 1). Indeed, because traits represent consistent functional strategies, ‘slow’ and ‘fast’ traits are expected to covary. This guild-level aggregation from individual traits CWM to group-level slow-fast axes was necessary to make the groups comparable in the following analyses. Also, this approach was chosen because the main objective of the study was not to establish the existence of such guild-level slow-fast axes, but rather examine their association at the whole community level.”

See also our response R5.1a below.

Extended data figure 1: Identification of slow-fast axes for each guild. PCA were run on the CWM of traits hypothesized to be related to the slow-fast strategies in each guild to identify a common slow-fast axis (green arrows). This was the first PC axis for all guilds, except aboveground secondary consumer arthropods. The evidence supported the hypothesized slow-fast axis for most groups, and partially supported it for three groups. For Lepidoptera, there was no response from body size, contrary to expectations. Body size was measured as wing length, an indicator of overall body size (which is expected to be a “slow” trait; smaller size helps to

survive disturbance) but also of dispersal ability (a “fast” trait, as larger wings promote recolonisation after disturbance). Both effects might cancel each other leading to no response of size. For secondary consumer, above-ground arthropods, dispersal and body size are slightly confounded because larger body size increases dispersal abilities. However, these two traits are opposed on the second axis of the PCA, which we use as our slow-fast index. The last group for which we found only partial support for the slow-fast axis was bats. High body mass is usually considered a ‘slow’ trait, and is expected to be positively correlated to lifespan and negatively to the number of offspring (trade-off between survival and reproduction). However, hibernation saves resources and leads, in hibernating bats (most of the species observed in our study), to a correlation between number of offspring and body mass (Wilkinson 2002), leading to the results observed here.

Extended data figure 2. The whole community level slow-fast axis, based on all traits. In contrast to the results shown in Figure 2, where the PCA was conducted on guild-level slow-fast axes, the PCA was conducted on all traits, at the CWM level. Each trait was weighted as $1/n$, with n the number of traits available for the group, so that all groups are weighted equally. Traits expected to be “slow” are coded in blue, “fast” in green.

4.6. Ultimately, while I support the endeavour of this manuscript, I felt real mechanism was

missing linking fast-slow traits to fast-slow ecosystem system processes, and the discussion was too general in nature to be useful.

R4.6 We conducted some additional analyses to evaluate whether individual traits or groups were more specifically associated to individual functions (see response R5.4 below). We have also expanded the text on how fast-slow traits translate into fast-slow ecosystem functions in both the introduction and discussion:

I. 175: “Finally, many studies have shown that community-level trait measures of individual guilds explain variation in individual ecosystem functions, which is to be expected given the link between functional traits and rates of metabolic and tissue turnover processes (Buzzard et al., 2019; Cadotte, 2017; Evans et al., 2017)”

I. 527. “Finally, we provide evidence that this whole-community strategy variation mediates the effect of land use intensity on overall ecosystem functioning. More specifically, communities dominated by faster, non-conservative strategies (e.g. faster reproduction, shorter lifespan) have a faster metabolic rates per unit biomass, a higher digestibility and/or tissue resource concentrations and rapid turnover of tissues. All these factors lead to a faster transfer of resources from organic to inorganic pools and between trophic levels (Wardle et al., 2004, Chapin et al. 2011, Yvon-Durocher et al. 2012). In terms of ecosystem functioning this equates to greater productivity and gas fluxes, higher soil enzyme activities, and faster decomposition and rates of organic matter mineralization. This extends previous findings that demonstrate the linkages between traits of individual guilds and ecosystem functions (Buzzard et al., 2019; Cadotte, 2017; Evans et al., 2017).”

Thank you for your constructive feedback.

Reviewer #5 (Remarks to the Author):

This paper investigates the effect of land use intensity on community-level mean values of traits that reflect a continuum of functional strategies from slow- to fast growing organisms. The authors analyzed an impressive occurrence and trait database and identified the main slow-fast functional trait axes in 14 trophic guilds. They show direct and indirect effects of land use intensity on the slow-fast trait axes. They also showed that the slow-fast trait axes were associated with the rates of whole ecosystem functioning. The authors conclude by discussing the potential of the observed ‘slow-fast’ continuum for whole ecosystem-level studies.

I very much enjoyed reading this manuscript which has a clear structure and is well written. The topic of the study fits within the scope of the journal. The main questions are relevant and timely. Overall, the work and methods meet the expected scientific standards. The conclusions are supported by the results. The figures and tables are clear and of good quality. The literature cited is appropriate.

The study summarizes complex processes in a simple message that can help improve future ecosystem conservation and management efforts. I think the study will be of interest to a broad range of researchers and practitioners in the field of ecology and beyond. I have not found any major weaknesses or errors in this study. However, I make several suggestions for possible minor improvements.

Thank you for your supportive and constructive feedback!

Note that, as requested, my review mostly focuses on the data analysis part of the manuscript. See my comments below for further details.

Main comments:

5.1. Trait selection

Although the authors did considerable efforts to explain this process, I think that the interpretation of PCA axes as slow-fast and their selection for further analyses can be seen as a weak point. For example, I agree with R3 that important traits in bats and birds are missing and would be easy to add given the high amount and quality of trait information for these two taxa. Furthermore, if traits were originally selected to represent the slow-fast continuum, why select just one PCA axis? The non-selected axes might represent other important ecological strategies. Traits related to reproduction might have different drivers than those related to growth. Some aspects of the slow-fast trait continuum might not be driven by land use intensity. To me, this selection process, especially, the selection of PCA axes, casts some doubts on the validity and generality of the conclusions.

However, the authors did provide a complementary analysis based on all traits (Fig.S3) where the slow-fast axis emerges in a more direct and organic way. This additional analysis alongside Fig.3 did put my initial concerns at ease as the results show that the slow-fast axis is present even without selecting PCA axes. I encourage the authors to put more emphasis on these two analyses in the text. For example, the authors could mention Fig.S3 directly in the caption of Fig.2. See also my comments about fourth corner/RLQ below.

R5.1a. Thank you for your suggestion. We agree that the process of defining group-level slow-fast axes is a bit circular. However, we would like to stress that the main objective of the paper was not to demonstrate the existence of slow-fast axes within individual groups, but to show that these axes were associated at the level of the whole community. For each group, we selected the PCA axis which best corresponded to our understanding of the fast-slow (even if was second, though it usually was the first). This made their inclusion in the next, comparative, steps of analysis possible, especially for the structural equation modelling (SEM).

Following your feedback, we considered replacing Fig. 2 with the previous Fig S3 as it shows the whole community slow-fast axis in a more intuitive way. We eventually decided against this as the group-level slow-fast axes are used in the SEM, and we expect that introducing them early on in the paper (Fig 2) will help readers to understand the analyses that follow. However, we now put more emphasis on the 'all traits' analysis in the text and by presenting the PCA with all traits as an extended data figure, making it much more accessible to readers than it was in the supplementary material.

I. 324: "This conclusion was supported by the analysis of individual traits' responses to land use intensity, which shows that the CWMs of 54% of individual traits responded to land use intensification in the hypothesised direction, and only two (4%) responding in an opposing direction (Figure 3)."

Figure 2. The slow-fast trait axis of individual guilds is strongly related to land-use intensity. The variables included in the PCA are the slow-fast axes of each guild, which were estimated as the PC axis that best represents a slow-fast axis (PC1 in 90% of cases) in a in a PCA of the selected CWM traits expected to be related to slow-fast strategies for each guild individually. These guild-level PCA can be found in Extended data Figure 1. Land-use intensity (which is projected on the PC axes, shown in red) is strongly associated with Axis 1.

Belowground guilds are shown in brown, aboveground guilds in blue, and plants in green. Sample size: 150 (sample sizes for each individual group are shown in Fig. 3, missing values were imputed to run the PCA, see methods). The corresponding analysis conducted on all traits, without guild-level aggregation, is shown as Extended data Figure 2.

R5.1b Regarding trait selection, we think that some traits suggested by Reviewer 3 (iteroparity, survivorship type) not clearly linked to slow-fast functional strategies and thus were not retained in our original screening of suitable traits. Reviewer 3 didn't raise this issue again so we hope our decision not to include them was appropriate.

5.2. Controlling for the effect of environmental covariates

Controlling for the effect of environmental covariates is indeed a necessary step and I agree with the approach used by the authors to address this issue. However, I miss summary information about this step. How much variation is explained in each taxon? Do we have large differences among taxa? Which variables explain the most variation? After my first read, I was wondering whether this step can cause some biases. However, Table S14 shows correlations between LUI and CWM where CWMs were not corrected for environmental covariates and the results are consistent with the other analyses. This suggests that the results are robust. I think that RLQ would be interesting to characterize the links between traits and environmental variables. This analysis would provide important information about the strength of the effect of LUI on traits as compared to that of other covariates. It would also provide a more general picture of the ecology of the different taxonomic groups. Whether or not the authors decide to use the RLQ, I think that more information is needed about the relative importance of the influence of LUI on traits as compared to that of other environmental variables.

Dray, S., Choler, P., Dolédec, S., Peres-Neto, P. R., Thuiller, W., Pavoine, S., & ter Braak, C. J. (2014). Combining the fourth-corner and the RLQ methods for assessing trait responses to environmental variation. *Ecology*, 95(1), 14-21. DOI 10.1890/13-0196.1

R5.2 We agree that more information on the response of individual groups and traits to the environmental variables considered would add transparency to the paper.

After careful consideration decided against using RLQ for a range of reasons. First, RLQ links traits of individual species to environmental covariates through the abundance of individual species. Our whole analysis is based on the expectation that community-level metrics are the appropriate scale to address our research question. Second, some of our traits are calculated directly at the level of the community (e.g. fungal-bacterial ratio, % pathogen protists) and thus cannot be integrated into an RLQ analysis. To keep the paper understandable did not want to present different analyses for some traits compared than others, as this would further add to the complexity of the manuscript.

Instead, we decided to show the results as simple variance partitioning analyses (figure shown as Figure S1) and now mention in the methods for which groups the impact of environmental covariates was the strongest.

Figure S1 Variance partitioning of each trait CWM (before environmental correction) between land use intensity and all the environmental covariates considered. Proportion of variance explained by each variable is calculated as the proportion of sum of squares associated to each variable in a linear model, with the trait CWM as response variable and all others as explanatory variables.

The methods text now reads:

I. 767: “To gain a more reliable estimate of how CWM trait data was related to the hypothesised drivers, it was necessary to correct for environmental covariates before analysis. As such correction has been shown to produce biased parameter estimates in the presence of correlation between the environmental covariates (Freckleton, 2002), we identified highly correlated variables from our list of potential covariates (mean annual temperatures and precipitation, topographic wetness index, soil clay and sand content, pH, depth and region). We excluded soil depth and sand content which were both highly anticorrelated to soil clay content ($|r| > 0.72$); and precipitation which were highly anticorrelated with temperature ($r = -0.77$). We thus retained mean annual temperature, topographic wetness index, soil clay content and pH as well as the region as covariates. To investigate the importance of these factors relative to land-use intensity, our focus variable, we fitted linear models with each trait CWM as a response and all covariates and land use intensity as explanatory variables. This showed that on average the region explained as much variance as land use intensity (around 8%), followed by the Topographic Wetness Index (2%), but that there was high variability across groups. For instance, Bats and Collembola were primarily driven by the region, secondary consumer protists by soil pH, and plants or above-ground primary consumer arthropods by land-use intensity (Figure S1). After that, we fitted linear models with each trait CWM as a response, and the covariates, excluding land use intensity, as explanatory variables. The residuals of these linear models were used for all further analyses.”

5.3. Stacking of models

I agree with the main logic behind the multiple PCAs. Basically, it is the same idea as in a multiple factor analysis, MFA (see refs below). It is, to me, a valid and interesting way to conduct a multi-taxa analysis. However, I also agree with R2 that the analysis procedure is relatively complex which makes it difficult to keep track of the potential biases and information loss at each step. All in all, I see no major problem with the stacking of models and encourage the authors to cite the MFA papers as a reference framework for their approach.

Escofier, B. and Pages, J. (1994) Multiple Factor Analysis (AFMULT package). Computational Statistics and Data Analysis, 18, 121-140.

Becue-Bertaut, M. and Pages, J. (2008) Multiple factor analysis and clustering of a mixture of quantitative, categorical and frequency data. Computational Statistic and Data Analysis, 52, 3255-3268.

Thank you for this suggestion, we now cite these papers as an additional justification in the methods section.

I. 834: “To test whether the slow-fast responses were synchronous across guilds, we ran a PCA on the previously identified slow-fast axes of all guilds, and projected the land-use intensity index (LUI) as a supplementary variable on this PCA. This approach of combining multiple PCAs into a “main” PCA follows the same logic as Multiple Factor Analysis (MFA (Bécue-Bertaut and Pagès, 2008; Escofier and Pagès, 1994)) and allowed us to simultaneously analyse the response of groups with different traits and individual responses. For this analysis only, missing values for the slow-fast axis at the plot level (shown at the individual trait level in Table S1) were given the average of the considered guild across all plots to allow comparison across guilds (imputed values: 1.8% of all values, 0-12% range within groups).”

5.4. Trait-function relationship

Although not essential to the story, I miss the direct relationship between traits and EF. Here, the fourth-corner/RLQ analysis could also be used to test the associations between individual traits and EF variables. I think that such analyses would constitute a complementary and more direct approach to highlight the slow-fast trait continuum. I expect that this analysis will reveal significant correlations that will further validate the other analyses and increase the confidence in the results.

R5.4 See our response R5.2 regarding why we think the use of RLQ analyses might not be appropriate in this context – our hypotheses relate to the whole system trait measures and ecosystem functions, rather than individual trait-ecosystem function relationships for which there is a large existing body of literature (some of which is cited) Further, RLQ cannot be used on all our trait measures. We also wanted to keep additional analyses simple, because as noted by Reviewer 4 the long supplementary materials already make the paper complex.

Instead, and so readers can assess which traits may be driving the higher-level trend seen in whole ecosystem properties, we opted to show two correlation matrices, of all traits x all functions and of individual groups slow-fast axes x functions bundles (the bundles were used previously to downweigh highly related functions, see methods). For space reasons, only the trait-functions correlations (not traits x traits or functions x functions) are shown.

We would like to thank Reviewer 5 for suggesting this new analysis which revealed an important pattern in our study. The analysis highlighted the importance of fungal:bacterial ratio, specifically, in driving the ecosystem functioning slow-fast axis. We think this is an important point which we now mention in both the results and discussion.

I. 438: “This whole ecosystem functioning slow-fast axis was better explained by the entire community ‘slow-fast’ traits axis ($r = 0.4$, $R^2 = 0.33$, $p < 10^{-6}$) than by other hypothesised drivers of ecosystem functioning (single guild community traits measures (plants and microbes), land-use intensity or taxonomic diversity, R^2 - 0.11 to 0.26, Table S5). The exception to this was fungal-bacterial ratio, which was correlated with the whole slow-fast axis ($r = -0.52$, $P < 10^{-6}$) and better explained the functions slow-fast axis ($R^2 = 0.47$, Figure S5, S6). This was likely due to the prevalence of soil-related measures in our selected set of functions. Additionally, some individual functions were more strongly associated with specific groups or traits (e.g. fast plant community traits were positively associated with biomass production (Figure S5).”

L. 587: “The close relationship between our ecosystem slow-fast axis and fungal-bacterial ratio also suggest that this axis is compatible with earlier literature pointing to a slow-fast functioning gradient from fungal to bacteria-dominated communities (Bardgett et al., 1996; de Vries et al., 2006; Wardle et al 2004) and that fungal to bacterial ratios could act as an indicator variable for the whole ecosystem slow-fast continuum.”

Figure S 1. Pearson correlations between individual traits CWM (and overall community slow-fast trait axis) and functions (and overall slow-fast functioning axis). Both functions and traits were corrected for the environment beforehand.

Figure S 2. Pearson correlations between individual guild slow-fast axis (and overall community slow-fast trait axis) and functions bundles (and overall slow-fast functioning axis). Both functions and traits were corrected for the environment beforehand.

Minor comments:

5.5 The use of CWM: Whether CWM reflects optimal strategies has been subject to much discussion. I personally think that this is not always the case. For instance, a bimodal distribution of trait values within a community can indicate the coexistence of two successful strategies. In such a case, the mean value would reflect a sub-optimal strategy. Having said that, I do not think that such bimodal trait distributions are highly frequent and this should not be an issue for the validity of the results. I, nevertheless, encourage the authors to add a few words about this.

R5.5 We have now revised this paragraph in response to Reviewer 4 and do not mention optimal strategies in the new version.

I. 115: “At the level of guilds and communities, winning strategies can be seen as manifesting as an emergent property, and represented in community-level trait measures, typically the community abundance weighted trait mean (Bruehlheide et al., 2018; Garnier et al., 2004) (CWM). While there can be significant trait variation within a community, a CWM captures the average functional strategy of the community; and changes in CWM across space and time reflect both turnover in species with different trait values, and variation in their relative abundance, in response to changes to the species pool and environmental conditions. The combined response of multiple trait CWMs thus represents a change in the overall functional strategy at a community level, and this joint response is expected to be stronger than that of univariate traits (Muscarella and Uriarte, 2016). This means that slow-fast strategy responses – encompassing a range of traits – may emerge at the community level from a concurrent change in individual CWM traits related to ‘slow’ and ‘fast’ strategies. In particular, communities

of resource-acquisitive, fast-growing organisms with numerous offspring, a fast pace of life and good dispersal abilities tend to be found in resource-rich and disturbed habitats, while resource-conservative, slow-growing organisms with longer life span and fewer offspring tend to be favoured in undisturbed or resource-poor habitats (Börschig et al., 2013; Daou et al., 2021; de Vries et al., 2006; Pedley and Dolman, 2014; Simons et al., 2016) »

5.6 It was difficult for me to understand how was Fig.3 done. I encourage the authors to provide a few additional explanations in the text and directly in the caption.

We detailed the caption, and the results and methods now point more explicitly to Figure 3.

Figure 3. Observed responses of individual traits at the guild level (CWM) to land-use intensity, shown as estimated parameter +/- 95% confidence intervals. The responses shown were extracted for each trait individually as the slope +/- confidence interval of a linear model with the CWM trait as a response to land use intensity after correction for other environmental covariates (see Fig. S1) P-values were adjusted for multiple testing. [...]

I. 322: “The observed covariation in slow-fast traits across guilds demonstrates that land-use intensity drives differentiation between entire communities, with ‘slow’ and ‘fast’ communities characterised by distinct trait syndromes (Figure 2, Table S1). This conclusion was supported by the analysis of individual traits’ responses to land use intensity, which shows that the CWMs of 54% of individual traits responded to land use intensification in the hypothesised direction, and only two (4%) responding in an opposing direction (Figure 3)”

I. 803: “To test for the response of each individual CWM trait to land-use intensity, we calculated the slope of the regression of the CWM trait against land use intensity (excluding NAs) between each trait and land-use intensity (p-values corrected for false detection rates using the p.adjust function, n = 47, shown in Figure 3).”

5.7 Figure 4: Having two slightly different color scales for model estimates is confusing. Would it be possible to use one single scale for both panels?

Done, thanks for the good suggestion.

5.7 Table S6: What is the shared variation?

R5.7 We have now added the shared variance to Table S6 as requested:

Table S 5. Variance partitioning of land use intensity and multivariate trait community weighted mean across sampling plot and years. The partitioning was done using the varpart function (package vegan). Only groups with more than one sampling year are included. For bacteria and fungi, year 2017 was excluded because only one trait was available (% pathogen fungi) which did not allow us to properly partition the variance. Negative shared variance (a common artifact when using varpart) are shown as 0* (or <0.01* if between 0 and -0.01).

Variable	Variance attributed to the plot only	Variance attributed to the year only	Shared variance between year and plot	Residuals
LUI	0.70	<0.01	0*	0.30

Variable	Variance attributed to the plot only	Variance attributed to the year only	Shared variance between year and plot	Residuals
Arthropods (above-	0.09	0.01	0	0.90

					ground, secondary consumers)				
Birds	0.25	0.18	0.02	0.60	Arthropods (above-ground, primary consumers)	0.23	0.04	<0.01*	0.72
Bats	0.43	<0.01	<0.01	0.57	Bacteria and fungi	0.68	0.03	0*	0.24
Plants	0.52	0.01	<0.01*	0.47	Protists (secondary consumers)	0.22	<0.01	<0.01	0.78
Arthropods (below-ground, secondary consumers)	0.10	0.03	<0.01	0.87	Protists (bacterivores)	0.15	0.08	0*	0.68
Arthropods (below-ground, primary consumers)	0.17	0.06	0.02	0.79	Protists (plant pathogens)	0.38	0.15	0*	0.3

Overall, this is a very nice work and I hope that my comments will help to improve it further.

Thank you for your constructive review and useful suggestions.

Reviewers' Comments:

Reviewer #4:

Remarks to the Author:

The fast-slow spectrum is well recognized, and rarely synthesised across both aboveground and belowground components of ecosystems. However, inspecting the data (as far as I can tell from supplementary information) there is very little sampling and utility of the belowground aspects of the data analysis. It appears there is a single sampling event with low replication (4 samples per plot - unclear how many plots and what the treatments were). It is also unclear if all the data are from a single experiment or represent a larger dataset of multiple experiments. I think the original data and treatment of that data needs to be more transparent.

The authors state that 'Full code to replicate the analyses is stored on GitHub and will be made openly available before publication'. I would argue this is necessary for a reviewer to follow along on what the authors have done.

I really struggle to understand what the different figures and results actual represent because there are so many data manipulations performed.

At the same time, in the reply to reviewer comments letter, the authors have done a good and considerate job addressing comments and discussing key points. Specifically, the authors have removed the comments about 'optimality', clarified the CWM discussion, and changed the caption for Figure 2.

However, my concern for this whole manuscript is data transparency - I'm not sure someone could recreate this based on the multitude of data manipulation and transformations the authors have performed, and the information on guild traits, and other methods used to generate these results. There is still (as another reviewer also suggested) something seeming highly circular in this entire endeavour.

Please also check and confirm PCA vs PCoA.

Check appropriateness of references (e.g. M&W 1967 in regard to protist cell size).

Gamasidae is not an oribatid mite family (or any mite family)

267-291 - clarify belowground arthropod guilds. Are they omnivores, primary consumers, secondary consumers?

Reviewer #5:

Remarks to the Author:

I have read through and carefully considered the revised manuscript, with special attention toward the concerns previously raised by myself and the other reviewers.

I am satisfied with the authors' responses and the revisions to the manuscript. In particular, I appreciate the major effort made to improve the statistical analyses. I also appreciate the addition of Figure S2, which confirms and strengthens the previous results. I think these changes have strengthened the study.

Main comments:

1. Trait selection

OK, but pity to let the figure in the SupMat... To me, this figure really justifies the scale of your approach and sends a strong and important message to the readers. Fig.2 does a similar job but in a less intuitive way. All in all, I'm fine with the chosen approach and appreciate that Extended data figure 2 is now mentioned in the caption of Figure 2.

2. Controlling for the effect of environmental covariates

Fig.S1: Nice and very useful! Consider adding a label to the x-axis.

3. Stacking of models

OK

4. Trait-function relationship

I actually think that a 4th corner analysis would be complementary in that it could show the role of individual trait-environment relationship for the whole system which would provide useful information about the role and relative importance of processes occurring at different scales. Nevertheless, I'm happy with the addition of the two correlations matrices and generally agree with the argument of keeping the paper simple.

The correlation between the fungal:bacterial ratio and the ecosystem functioning slow-fast axis makes a strong link to previous work in the field. I'm happy that my comment has been useful.

5 CWM reflects optimal strategies

OK

6 Fig.3

OK, but I miss the R² of the different models. How much variation in CWM is explained by LUI? Consider adding a statement along the lines of "LUI explained between XX and YY % of the total variation in the different CWM (see figure S1)" in the caption.

Dear reviewers,

Thank you for your feedback on the previous version of our manuscript. We have now integrated these modifications: in particular, following the requests of the Editor and Reviewer 4, we have now moved all the data collection details to the methods' main text and clarified when needed the number of plots and sampling years for all groups. We have answered the other specific editorial guidelines in the attached Checklist.

On the behalf of all authors,

Margot Neyret

REVIEWERS' COMMENTS

Reviewer #4 (Remarks to the Author):

The fast-slow spectrum is well recognized, and rarely synthesised across both aboveground and belowground components of ecosystems. However, inspecting the data (as far as I can tell from supplementary information) there is very little sampling and utility of the belowground aspects of the data analysis. It appears there is a single sampling event with low replication (4 samples per plot - unclear how many plots and what the treatments were). It is also unclear if all the data are from a single experiment or represent a larger dataset of multiple experiments. I think the original data and treatment of that data needs to be more transparent.

We have now moved the methods tables, including information on sampling methodology, to the main text and clearly state the number of plots sampled for each group. There was no specific treatment/experiment; 150 plots were initially chosen along a land use intensity gradient and sampling was conducted in the same plots over several years. We state:

l. 535: We sampled vascular plants in an area of 4 m × 4 m in all 150 plots, and estimated the percentage cover of each occurring species every year from 2008 to 2019

l. 548: For bacteria the analysis only included data from 2011 (148 plots) and 2014 (150 plots)

l. 558: Data was available for 150 plots in 2011, 2014 and 2017.

l. 566: Data for one plot in 2011 was missing, data for all 150 plots was available in 2017

l. 580: Collembola data was available for 140 plots. For Oribatid mites, it was available in 149 plots but some plots had to be excluded due to low trait data coverage (see below), resulting in 136 used plots in total

l. 585: Data was available for 137 plots

I. 590: All arthropods of the herb layer were sampled twice per year between 2008 and 2017 in June and August to represent different phenological windows within the peak season of adult arthropod activity. Total number of plots sampled each year varied from 143 to 150, all plots were sampled at least 8 years (average: 9.8 years).

The authors state that ‘Full code to replicate the analyses is stored on GitHub and will be made openly available before publication’. I would argue this is necessary for a reviewer to follow along on what the authors have done.

Thank you. We now cite the code doi in the code availability statement.

I. 1035: Full code to replicate the analyses is stored on GitHub at https://github.com/mneyret/trait_synchronies under the DOI 10.5281/zenodo.10286643.

I really struggle to understand what the different figures and results actual represent because there are so many data manipulations performed.

We realize the paper is complex but have worked hard to make the data analysis performed as clear as possible, and the results intuitive and accessible.

At the same time, in the reply to reviewer comments letter, the authors have done a good and considerate job addressing comments and discussing key points. Specifically, the authors have removed the comments about ‘optimality’, clarified the CWM discussion, and changed the caption for Figure 2.

However, my concern for this whole manuscript is data transparency - I’m not sure someone could recreate this based on the multitude of data manipulation and transformations the authors have performed, and the information on guild traits, and other methods used to generate these results. There is still (as another reviewer also suggested) something seeming highly circular in this entire endeavour.

The code provided gives full transparency and allows for repeatability. The code includes several scripts which cover all the steps of the analysis, including the matching of trait and abundance data (with the links to individual datasets and a statement on whether they are currently available or not). We also provide the main script of the analysis, which starts from the CWM and environmental dataset to the production of all results and figures. All datasets are cited in the relevant paragraph in the methods, and this is also detailed in the data availability statement:

L. 1018: The Community Weighted Mean data generated in this study have been deposited under accession code 31516 <https://www.bexis.uni-jena.de/ddm/data/Showdata/31516>. The raw trait and abundance, and ecosystem functions datasets are detailed below, many of which are publicly available. To give data owners and collectors time to perform their analysis the Biodiversity Exploratories' data and publication policy includes by default an embargo period of three years from the end of data collection/data assembly. Both the CWM and remaining raw data datasets are

thus available under restricted access, access can be obtained by contacting data owners (listed on Bexis). At the end of the embargo period these datasets will be made publicly available via the same data repository.

Full list of used datasets (both from the Biodiversity Exploratories and external, previously published datasets): ^{3,19,45,47,53–56,93–97,100–117,219–233,237,245–250,254–263,273–275}.

Please also check and confirm PCA vs PCoA.

PCA was used at all steps.

Check appropriateness of references (e.g. M&W 1967 in regard to protist cell size).

M&W 1967 is cited in Table S1 as a general reference of why protist cell size is a relevant trait for this study (because body size in general is relevant to fast-slow strategies), not to justify the data used for cell size.

Gamasidae is not an oribatid mite family (or any mite family)

Thanks for the correction; there was a small mistake in the dataset which we now corrected. This had no impact on the results (except in terms of trait coverage) as this group were previously considered as missing trait data.

267-291 - clarify belowground arthropod guilds. Are they omnivores, primary consumers, secondary consumers?

Both primary consumers and secondary consumer arthropods are included. The trophic group considered is indicated in parenthesis after, e.g. belowground arthropods (secondary consumers)

Reviewer #5 (Remarks to the Author):

I have read through and carefully considered the revised manuscript, with special attention toward the concerns previously raised by myself and the other reviewers.

I am satisfied with the authors' responses and the revisions to the manuscript. In particular, I appreciate the major effort made to improve the statistical analyses. I also appreciate the addition of Figure S2, which confirms and strengthens the previous results. I think these changes have strengthened the study.

Thank you for your feedback and positive assessment.

Main comments:

1. Trait selection

OK, but pity to let the figure in the SupMat... To me, this figure really justifies the scale of your approach and sends a strong and important message to the readers. Fig.2 does a similar job but in a less intuitive way. All in all, I'm fine with the chosen approach and appreciate that Extended data figure 2 is now mentioned in the caption of Figure 2.

Thank you. Putting this figure as extended data, rather than in the supplementary material, will make it visible to most readers of the article as it will link directly from the article main page.

2. Controlling for the effect of environmental covariates

Fig.S1: Nice and very useful! Consider adding a label to the x-axis.

Thank you, we now added a label for each functional group of the figure:

3. Stacking of models

OK

4. Trait-function relationship

I actually think that a 4th corner analysis would be complementary in that it could show the role of individual trait-environment relationship for the whole system which would provide useful information about the role and relative importance of processes occurring at different scales. Nevertheless, I'm happy with the addition of the two correlations matrices and generally agree with the argument of keeping the paper simple.

The correlation between the fungal:bacterial ratio and the ecosystem functioning slow-fast axis makes a strong link to previous work in the field. I'm happy that my comment has been useful.

It was- thank you for your contribution!

5 CWM reflects optimal strategies

OK

6 Fig.3

OK, but I miss the R2 of the different models. How much variation in CWM is explained by LUI? Consider adding a statement along the lines of "LUI explained between XX and YY % of the total variation in the different CWM (see figure S1)" in the caption.

Thank you for your suggestion. We now added the following sentence to the legend of Figure 3. Note that we reference Table S2 rather than Figure S1 as Fig S1 shows non-environmentally corrected CWM which might lead to different R2.

Figure 3: [...] LUI explained between 0 (*non-significant traits*) and 40% (*Lepidoptera voltinism*) of the total variation in individual trait CWM (Table S2). [...]